# Robust estimation of cancer and immune cell-type proportions from bulk tumor ATAC-Seq data

Aurélie Anne-Gaëlle Gabriel[1,2,3,4], Julien Racle[1,2,3,4], Maryline Falquet[3,5,6,7], Camilla Jandus[3,5,6,7], David Gfeller[1,2,3,4]*

[1]Department of Oncology, Ludwig Institute for Cancer Research, University of Lausanne, Lausanne, Switzerland; [2]Agora Cancer Research Center, Lausanne, Switzerland; [3]Swiss Cancer Center Leman (SCCL), Geneva, Switzerland; [4]Swiss Institute of Bioinformatics (SIB), Lausanne, Switzerland; [5]Ludwig Institute for Cancer Research, Lausanne Branch, Lausanne, Switzerland; [6]Department of Pathology and Immunology, Faculty of Medicine, University of Geneva, Geneva, Switzerland; [7]Geneva Center for Inflammation Research, Geneva, Switzerland

*For correspondence:
david.gfeller@unil.ch

**Abstract** Assay for Transposase-Accessible Chromatin sequencing (ATAC-Seq) is a widely used technique to explore gene regulatory mechanisms. For most ATAC-Seq data from healthy and diseased tissues such as tumors, chromatin accessibility measurement represents a mixed signal from multiple cell types. In this work, we derive reliable chromatin accessibility marker peaks and reference profiles for most non-malignant cell types frequently observed in the microenvironment of human tumors. We then integrate these data into the EPIC deconvolution framework (Racle et al., 2017) to quantify cell-type heterogeneity in bulk ATAC-Seq data. Our EPIC-ATAC tool accurately predicts non-malignant and malignant cell fractions in tumor samples. When applied to a human breast cancer cohort, EPIC-ATAC accurately infers the immune contexture of the main breast cancer subtypes.

## eLife assessment

This study presents an **important** computational tool for the quantification of the cellular composition of human tissues profiled with ATAC-seq. The methodology and its application results on breast cancer tumor tissues are **convincing**. It advances existing methods by utilizing a comprehensive reference profile for major cancer-relevant cell types, compatible with a widely-used cell type deconvolution tool.

## Introduction

Gene regulation is a dynamic process largely determined by the physical access of chromatin-binding factors such as transcription factors (TFs) to regulatory regions of the DNA (e.g. enhancers and promoters; *Klemm et al., 2019*). The genome-wide landscape of chromatin accessibility is essential in the control of cellular identity and cell fate and thus varies in different cell types (*Klemm et al., 2019*; *Zhang et al., 2021b*). Over the last decade, Assay for Transposase-Accessible Chromatin (ATAC-Seq; *Buenrostro et al., 2013*) has become a reference epigenomic technique to profile chromatin accessibility and the activity of gene regulatory elements in diverse biological contexts including cancer (*Luo et al., 2022*) and across large cohorts (*Corces et al., 2018*). Several optimized ATAC-Seq protocols have been developed to improve the quality of ATAC-Seq data and expand its usage to different

tissue types. These include the OMNI-ATAC protocol, which leads to a cleaner signal and is applicable to frozen samples (*Corces et al., 2017*; *Grandi et al., 2022*), as well as the formalin-fixed paraffin-embedded (FFPE)-ATAC protocol adapted to FFPE samples (*Zhang et al., 2022b*). The reasonable cost and technical advantages of these protocols foreshadow an increased usage of ATAC-Seq in cancer studies (*Grandi et al., 2022*; *Luo et al., 2022*).

Most biological tissues are composed of multiple cell types. For instance, tumors are complex ecosystems including malignant and stromal cells as well as a large diversity of immune cells. This cellular heterogeneity impacts tumor progression and response to immunotherapy (*Fridman et al., 2012*; *Fridman et al., 2017*; *de Visser and Joyce, 2023*). Most existing ATAC-Seq data from tumors were performed on bulk samples, thereby mixing the signal from both cancer and non-malignant cells. Precisely quantifying the proportions of different cell types in such samples represents a promising way to explore the immune contexture and the composition of the tumor microenvironment (TME) across large cohorts. Carefully assessing cell-type heterogeneity is also important in handling confounding factors in genomic analyses in which samples with different cellular compositions are compared. Recently, single-cell ATAC-Seq (scATAC-Seq) has been developed to explore cellular heterogeneity with high resolution in complex biological systems (*Cusanovich et al., 2015*; *Lareau et al., 2019*; *Satpathy et al., 2019*). However, the resulting data are sensitive to technical noise and such experiments require important resources, which so far limits the use of scATAC-Seq in contrast to bulk ATAC-Seq in the context of large cohorts.

In the past decade, computational deconvolution tools have been developed to predict the proportion of diverse cell types from bulk genomic data obtained from tumor samples (*Becht et al., 2016*; *Racle et al., 2017*; *Avila Cobos et al., 2018*; *Avila Cobos et al., 2020*; *Finotello et al., 2019*; *Monaco et al., 2019*; *Newman et al., 2019*; *Sturm et al., 2019*; *Li et al., 2020b*). As described in more detail elsewhere (*Avila Cobos et al., 2018*; *Sturm et al., 2019*), many of these tools model bulk data as a mixture of reference profiles either coming from purified cell populations or inferred from single-cell genomic data for each cell type. The accuracy of cell-type proportion predictions relies on the quality of these reference profiles as well as on the specificity of cell-type markers (*Avila Cobos et al., 2018*). A limitation of most deconvolution algorithms is that they do not predict the proportion of cell types that are not present in the reference profiles (hereafter referred to as 'uncharacterized' cells). In the context of tumor samples, these uncharacterized cell populations include malignant cells whose molecular profiles differ not only from one cancer type to another but also from one patient to another even within the same tumor type (*Corces et al., 2018*). To address this limitation, a few tools developed for bulk RNA-Seq data consider uncharacterized cells in their deconvolution framework by using cell-type specific markers not expressed in the uncharacterized cells (*Gosink et al., 2007*; *Clarke et al., 2010*; *Racle et al., 2017*; *Finotello et al., 2019*). These tools include EPIC (estimating the proportion of immune and cancer cells) which simultaneously quantifies immune, stromal, vascular as well as uncharacterized cells from bulk tumor samples (*Racle et al., 2017*; *Racle and Gfeller, 2020*).

Most deconvolution algorithms have been developed for transcriptomic data (RNA-Seq data) (*Gong and Szustakowski, 2013*; *Newman et al., 2015*; *Newman et al., 2019*; *Racle et al., 2017*; *Finotello et al., 2019*; *Jiménez-Sánchez et al., 2019*; *Monaco et al., 2019*; *Li et al., 2020a*). More recently they have been proposed for other omics layers such as methylation (*Aryee et al., 2014*; *Teschendorff et al., 2017*; *Teschendorff et al., 2020*; *Chakravarthy et al., 2018*; *Hicks and Irizarry, 2019*; *Rahmani et al., 2019*; *Arneson et al., 2020*; *Zhang et al., 2021a*; *Salas et al., 2022*), proteomics (*Feng et al., 2023*), or chromatin accessibility (*Li et al., 2020b*). For the latter, a specific framework called DeconPeaker (*Li et al., 2020b*) was developed to estimate cell-type proportions from bulk ATAC-Seq samples. Deconvolution tools developed initially for RNA-Seq data can also be applied to ATAC-Seq. For example, the popular deconvolution tool, CIBERSORT (*Newman et al., 2015*), in combination with ATAC-Seq profiles, was used to deconvolve leukemic ATAC-Seq samples (*Corces et al., 2016*).

Other methods have been proposed to decompose ATAC-Seq bulk profiles into subpopulation-specific profiles (*Burdziak, 2019*; *Zeng et al., 2019*) or compartments (*Peng et al., 2019*). However, these methods have more requisites: (i) the integration of the ATAC-Seq data with single-cell or bulk RNA-Seq (*Burdziak, 2019*; *Zeng et al., 2019*) and HIChIP data (*Zeng et al., 2019*) or, (ii) subsequent feature annotation to associate compartments with cell types or biological processes (*Peng et al., 2019*).

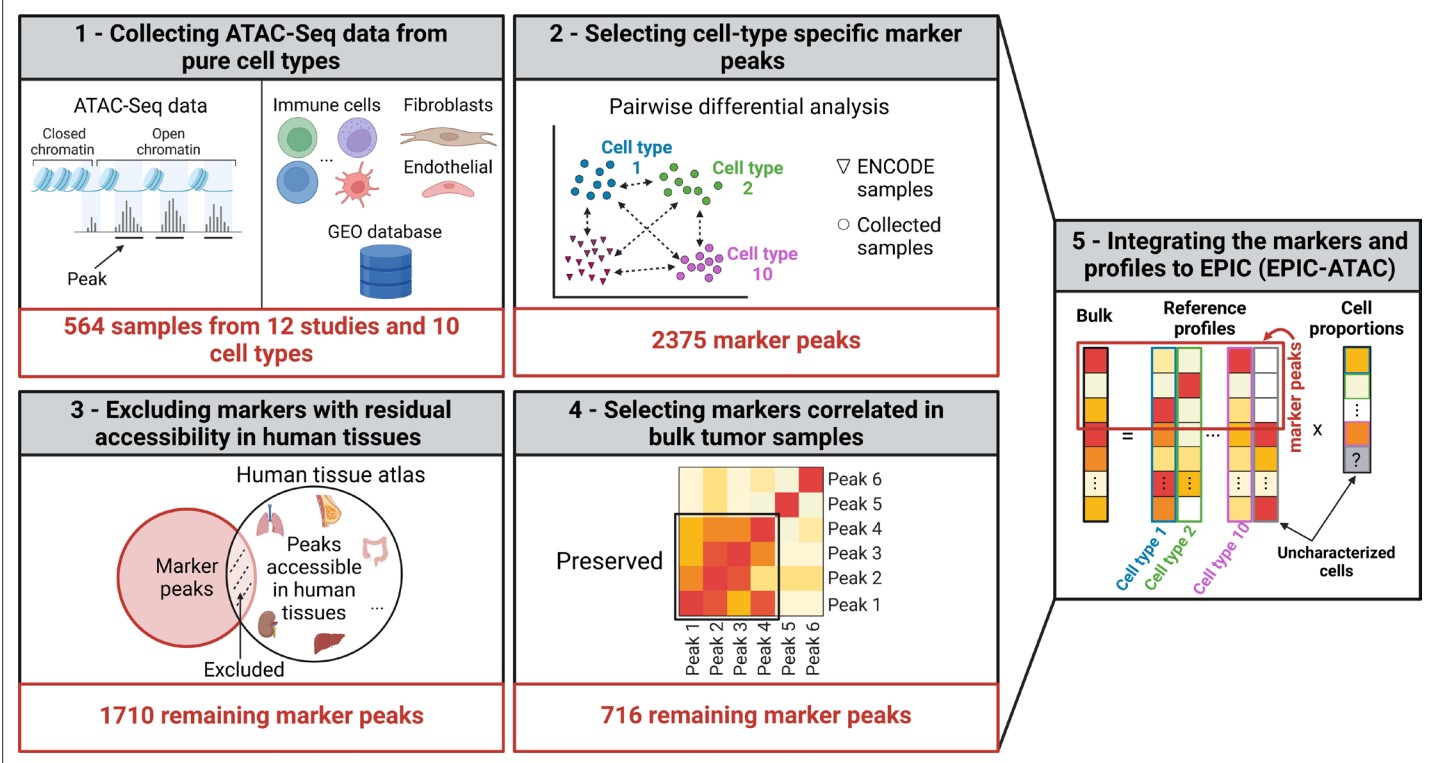

**Figure 1.** Graphical description of the identification of cell-type specific marker peaks and reference ATAC-Seq profiles included in the EPIC-ATAC framework. (1) 564 pure ATAC-Seq data of sorted cells were collected to build reference profiles for non-malignant cell types observed in the tumor microenvironment. (2) Cell-type specific marker peaks were identified using differential accessibility analysis. (3) Markers with previously observed chromatin accessibility in human healthy tissues were excluded. (4) For tumor bulk deconvolution, the set of remaining marker peaks was refined by selecting markers with correlated behavior in tumor bulk samples. (5) The cell-type specific marker peaks and reference profiles were integrated into the EPIC-ATAC framework to perform bulk ATAC-Seq deconvolution. Created with BioRender.com.

The online version of this article includes the following figure supplement(s) for figure 1:

**Figure supplement 1.** Chromatin accessibility of the markers in the reference and ENCODE samples.

The application of existing bulk ATAC-Seq data deconvolution tools to solid tumors faces some limitations. First, current computational frameworks do not quantify populations of uncharacterized cell types. Second, ATAC-Seq-based markers (i.e. chromatin-accessible regions called peaks) and reference profiles generated so far have been derived in the context of hematopoietic cell mixtures (*Corces et al., 2016*; *Li et al., 2020b*). Markers and profiles for major populations of the TME (e.g. stromal and vascular cells) are thus missing. While cell-type specific markers have been identified from scATAC-Seq data (*Zhang et al., 2021b*), not all TME-relevant cell types are covered (e.g. lack of scATAC-Seq data from neutrophils due to extracellular traps formation). Many of these markers have also not been curated to fulfill the requirements of tools such as EPIC to quantify uncharacterized cells (i.e. markers of a cell type should not be accessible in other human tissues).

In this study, we collected ATAC-Seq data from pure cell types to identify cell-type specific marker peaks and to build reference profiles from most major non-malignant cell types typically observed in tumors. These data were integrated into the EPIC (*Racle et al., 2017*) framework to perform bulk ATAC-Seq samples deconvolution (*Figure 1*). Applied on peripheral blood mononuclear cells (PBMCs) and tumor samples, the EPIC-ATAC framework showed accurate predictions of the proportions of non-malignant and malignant cells.

# Results

## ATAC-Seq data from sorted cell populations reveal cell-type specific marker peaks and reference profiles

A key determinant for accurate predictions of cell-type proportions by most deconvolution tools is the availability of reliable cell-type specific markers and reference profiles. To identify robust chromatin accessibility marker peaks of cell types observed in the tumor microenvironment, we collected 564 samples of sorted cell populations from twelve studies including B cells (*Corces et al., 2016*; *Calderon et al., 2019*; *Zhang et al., 2022a*), CD4+ T cells (*Corces et al., 2016*; *Mumbach et al., 2017*; *Liu et al., 2020*; *Giles et al., 2022*; *Zhang et al., 2022a*), CD8+ T cells (*Corces et al., 2016*; *Calderon et al., 2019*; *Liu et al., 2020*; *Giles et al., 2022*; *Zhang et al., 2022a*), natural killer (NK) cells (*Corces et al., 2016*; *Calderon et al., 2019*), dendritic cells (DCs) (mDCs and pDCs are grouped in this cell-type category) (*Calderon et al., 2019*; *Leylek et al., 2020*; *Liu et al., 2020*), macrophages (*Liu et al., 2020*; *Zhang et al., 2022a*), monocytes (*Corces et al., 2016*; *Calderon et al., 2019*; *Leylek et al., 2020*; *Trizzino et al., 2021*; *Zhang et al., 2022a*), neutrophils (*Perez et al., 2020*; *Ram-Mohan et al., 2021*), fibroblasts (*Liu et al., 2020*; *Ge et al., 2021*), and endothelial cells (*Liu et al., 2020*; *Xin et al., 2020*; *Figure 1* box 1, *Figure 2A*, *Supplementary file 1*). To limit batch effects, the collected samples were homogeneously processed from read alignment to peak calling. For each cell type, we derived a set of stable peaks observed across samples and studies, that is for each study, peaks detected in at least half of the samples were considered, and for each cell type, only peaks detected jointly in all studies were kept (see Materials and methods, 'Generation of an ATAC-Seq reference dataset of non-malignant cell types frequently observed in the tumor microenvironment').

These peaks were then used to perform pairwise differential analysis to identify marker peaks for each cell type (*Figure 1*, Box 2). To ensure that the cell-type specific marker peaks are not accessible in other human tissues, we included in the differential analysis ATAC-Seq samples from diverse human tissues from the ENCODE data (*Abascal et al., 2020*; *Rozowsky et al., 2023*; *Figure 1—figure supplement 1*). To select a sufficient number of peaks prior to peak filtering, the top 200 peaks recurrently differentially accessible across all cell-type pairs were selected as cell-type specific markers (see Materials and methods, 'Identification of cell-type specific markers'). To filter out markers that could be accessible in other human cell types than those included in our reference profiles, we used the human atlas study (*Zhang et al., 2021b*), which identified modules of open chromatin regions accessible in a comprehensive set of human tissues, and we excluded from our marker list the markers overlapping these modules (*Figure 1*, box 3, see Materials and methods 'Identification of cell-type specific markers'). The resulting marker peaks specific only to the immune cell types were considered for the deconvolution of PBMC samples (PBMC markers). For the deconvolution of tumor bulk samples, the lists of marker peaks specific to fibroblasts and endothelial cells were added to the PBMC markers. This extended set of markers was further refined based on the correlation patterns of the markers in tumor bulk samples from the diverse solid cancer types from The Cancer Genome Atlas (TCGA; https://www.cancergenomicscloud.org/datasets; *Corces et al., 2018*), that is markers exhibiting the highest correlation patterns in the tumor bulk samples were selected using the *findCorrelation* function from the caret R package (*Kuhn, 2008*; *Figure 1*, box 4, see the Materials and methods, 'Identification of cell-type specific markers'). The latter filtering ensures the relevance of the markers in the TME context since cell-type specific TME markers are expected to be correlated in tumor bulk ATAC-Seq measurements (*Qiu et al., 2021*). A total of 716 markers of immune, fibroblasts, and endothelial cell types remained after the last filtering (defined as TME markers). Considering the difference in cell types and the different filtering steps applied on the PBMC and TME markers, we recommend using the TME markers and profiles to deconvolve bulk samples from tumor samples and the PBMC markers and profiles to deconvolve PBMC samples.

To assess the quality and reproducibility of these markers, we first performed principal component analysis (PCA) based on each set of marker peaks. Computing silhouette coefficients based on the cell-type classification and on the study of origin showed that samples clustered by cell type and not by study of origin (averaged silhouette coefficients above 0.45 for cell type and around 0 for the study of origin). The representation of the samples based on the first axes of the PCA confirmed this observation (*Figure 2B* and *Figure 2—figure supplement 1*). These results indicate limited remaining batch effects after data processing and marker selection.

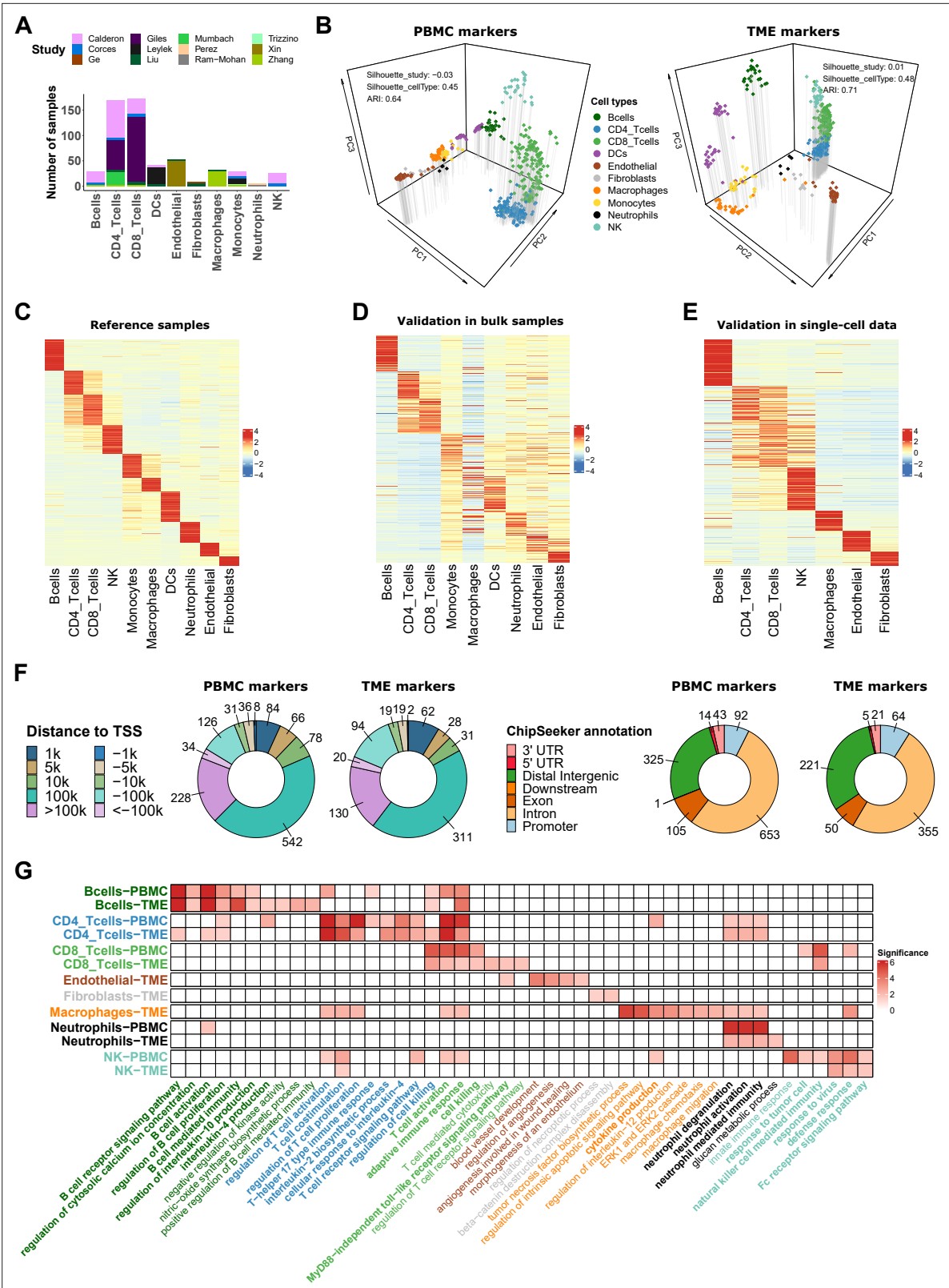

**Figure 2.** ATAC-Seq data from sorted cell populations reveal cell-type specific marker peaks and reference profiles. (**A**) Number of samples collected for each cell type. The colors correspond to the different studies of origin. (**B**) Representation of the collected samples using the first three components of the PCA based on the PBMC markers (left) and TME markers (right). Colors correspond to cell types. (**C**) Scaled averaged chromatin accessibility of all the cell-type specific marker peaks (PBMC and TME markers) (rows) in each cell type (columns) in the ATAC-Seq reference samples used to identify the

*Figure 2 continued on next page*

*Figure 2 continued*

marker peaks. (**D**) Scaled averaged chromatin accessibility of all the marker peaks in external ATAC-Seq data from samples of pure cell types excluded from the reference samples (see Materials and methods). (**E**) Scaled averaged chromatin accessibility of all the marker peaks in an external scATAC-Seq dataset (Human Atlas *Zhang et al., 2021b*). (**F**) Distribution of the marker peak distances to the nearest transcription start site (TSS) (left panel) and the ChIPseeker annotations (right panel). (**G**) Significance (-log10(q.value)) of pathways (columns) enrichment test obtained using ChIP-Enrich on each set of cell-type specific marker peaks (rows). A subset of relevant enriched pathways is represented. Colors of the names of the pathways correspond to cell types where the pathways were found to be enriched. When pathways were significantly enriched in more than one set of peaks, pathway names are written in bold.

The online version of this article includes the following figure supplement(s) for figure 2:

**Figure supplement 1.** Reference samples in the PCA space.

We then used the collected samples to generate chromatin accessibility profiles by computing the average of the normalized counts for each peak in each cell type as well as peak variability in each cell type (see Materials and methods) since EPIC uses features variability for the estimation of the cell-type proportions (*Racle et al., 2017*). *Figure 2C* represents the average chromatin accessibility of each marker peak in each cell type of the reference dataset and highlights, as expected, the cell-type specificity of the selected markers (see also *Supplementary files 2 and 3*), which was confirmed in independent ATAC-Seq data from sorted cells and single-cell ATAC-Seq samples from blood and diverse human tissues (*Figure 2D and E*, see Materials and methods).

## Annotations of the marker peaks highlight their biological relevance

To characterize the different marker peaks, we annotated them using ChIPseeker (*Yu et al., 2015*). We observed that most of the markers are in distal and intergenic regions (*Figure 2F*), which is expected considering the large proportion of distal regions in the human genome and the fact that such regions have been previously described as highly cell-type specific (*Corces et al., 2016*). We also noticed that 7% of the PBMC and TME marker peaks are in promoter regions in contrast to 4% when considering matched genomic regions randomly selected in the set of peaks identified prior to the differential analysis (see Materials and methods), which suggests enrichment in our marker peaks for important regulatory regions.

To assess the biological relevance of the marker peaks, we associated each marker peak with its nearest gene using ChIP-Enrich based on the 'nearest transcription start site (TSS)' locus definition (*Welch et al., 2014*; *Supplementary files 4 and 5*). The nearest genes reported as known marker genes in public databases of gene markers (i.e. PanglaoDB *Franzén et al., 2019* and CellMarker *Hu et al., 2023*) are listed in *Table 1*.

In each set of cell-type specific peaks, we observed an overrepresentation of chromatin binding proteins (CBPs) reported in the JASPAR2022 database (*Castro-Mondragon et al., 2022*) (using Signac *Stuart et al., 2021* and MonaLisa *Machlab et al., 2022* for assessing the overrepresentation) and the ReMap catalog (*Hammal et al., 2022*) (using RemapEnrich, see Materials and methods). Overrepresented CBPs also reported as known marker genes in the PanglaoDB and CellMarker databases are listed in *Table 1*. Detailed peak annotations are summarized in *Supplementary files 4 and 5*.

Based on the 'nearest TSS' annotation, we tested, using ChIP-Enrich (*Welch et al., 2014*), whether each set of cell-type specific marker peaks was enriched for regions linked to specific biological pathways (GO pathways). *Figure 2G* highlights a subset of the enriched pathways that are consistent with prior knowledge of each cell type. Some of these pathways are known to be characteristic of immune responses to inflammatory or tumoral environments. The complete list of enriched pathways is listed in the *Supplementary files 6 and 7*. Overall, these analyses demonstrate that the proposed cell-type specific marker peaks capture some of the known biological properties associated with each cell type.

## EPIC-ATAC integrates the marker peaks and profiles into EPIC to perform bulk ATAC-Seq data deconvolution

The cell-type specific marker peaks and profiles derived from the reference samples were integrated into the EPIC deconvolution tool (*Racle et al., 2017*; *Racle and Gfeller, 2020*). We will refer to this ATAC-Seq deconvolution framework as EPIC-ATAC. To ensure the compatibility of any input bulk ATAC-Seq dataset with the EPIC-ATAC marker peaks and reference profiles, we provide an option to lift over hg19 datasets to hg38 (using the liftOver R package) as the reference profiles are based on

**Table 1.** List of nearest genes and enriched CBPs reported in the PanglaoDB or CellMarker databases.

| Cell type | Nearest genes | Enriched CPBs |
|---|---|---|
| Bcells | DHTKD1 LHPP WDFY4 ARID5B HHEX SIDT2 CD82 MS4A1 FCHSD2 USP8 RHCG ATF7IP2 CIITA GGA2 SNX29P2 C16orf74 CBFA2T3 CD79B BCL2 GNG7 CD22 FCER2 FCRL1 LY9 PTPRC LAPTM5 IGLL5 VPREB3 CENPM AFF3 SP100 INPP5D DTNB CD86 RFTN1 ST6GAL1 NGLY1 OSBPL10 TLR9 CD38 SMIM14 ARHGAP24 ADAM19 EBF1 BASP1 CD83 PLEKHG1 CCR6 CCND3 HDAC9 CDCA7L BLK MTSS1 LYN PLEKHF2 MOB3B PAX5 | SPIB POU2F2 TCF4 EBF1 TCF3 NFKB1 STAT1 NFKB2 IKZF1 FOXO1 FOXP1 BCL6 POU2AF1 STAT3 BACH2 IKZF3 FLI1 TBX21 JUNB MITF NKX6-2 RBPJ |
| CD4_Tcells | IL2RA CD6 CD5 CD4 RORA PTPRC CTLA4 ICOS SLC9A9 FHIT TCF7 FYB1 ATXN1 CD40LG | TCF7 RUNX3 SOHLH2 IRF9 GATA3 TBX21 MAF RORA BATF CREM |
| CD8_Tcells | MKI67 JAML MAML2 KLRD1 NELL2 LAG3 PPP1R13B PTPRC LYST CASP8 CD8A CD8B CD96 BTLA GZMA THEMIS ETV1 | ETV1 FOXP3 TBX21 FOXP1 EOMES CREM IRF4 ZEB1 ARNT JUNB TCF7 |
| NK | PRF1 ZBTB16 KLRD1 SPN CD226 SH2D1B CD247 IL2RB CXCR4 NMUR1 GNLY ZAP70 TXK | EOMES TBX21 NFIL3 FOS JUN |
| DCs | C12orf75 LYZ APP CD8A RIOX2 NFKB1 QDPR ABCG2 PRELID2 DST CD36 IDO2 PCMTD1 | SPIB IRF8 MYB NR4A1 REL CUX2 FOXO1 ETV6 IRF5 BATF3 RUNX2 |
| Neutrophils | TLE3 CA4 CYP4F3 CEACAM8 PGLYRP1 FPR1 CTSS ALPL PI3 MMP9 CXCR1 DRC1 ASPRV1 LTF MGAM SLC25A37 | FOS |
| Monocytes | VENTX GLT1D1 CLEC4E CARS2 SLC24A4 C16orf74 FFAR2 STXBP2 NLRP3 CYRIA CMTM7 TGFBI DIAPH1 VCAN MCTP1 IFNGR1 STX11 CAPZA2 CD36 MTSS1 DENND3 ASAH1 TNFRSF10B BNIP3L NACC2 MAMDC2 FBP1 | CEBPA CEBPD CEBPB CEBPE SPI1 VENTX JUND RXRA TCF7L2 |
| Macrophages | CXCL12 PSAP P2RY6 SLCO2B1 CMKLR1 MMP19 LGMN CLEC10A C5AR1 FPR3 LILRB4 RGL1 SIGLEC1 MMP9 CD80 | STAT1 SPI1 FOSL2 FOS SPIC |
| Endothelial | FAM107B ROBO4 FLI1 ACVRL1 FLT1 DOCK9 ABCC1 S1PR1 ELOVL1 PLPP3 ASAP2 SNRK ECSCR ARAP3 LAMA4 BMP6 SERPINE1 LAMB1 DOCK4 NOS3 | ETV2 ELF1 FLI1 ELK3 FOSB ETS1 ERG GATA2 ZEB1 ETS2 FOXC1 SOX18 |
| Fibroblasts | LOX CAV1 COL15A1 | FOSL2 FOSB EGR1 FLI1 HIF1A PBX1 |

the hg38 reference genome. Subsequently, the features of the input bulk matrix are matched to our reference profiles' features. To match both sets of features, we determine for each peak of the input bulk matrix the distance to the nearest peak in the reference profiles peaks. Overlapping regions are retained and the feature IDs are matched to their associated nearest peaks. If multiple features are matched to the same reference peak, the counts are summed. Before the estimation of the cell-type proportions, we transform the data following an approach similar to the transcripts per million (TPM) transformation which has been shown to be appropriate for estimating cell fractions from bulk mixtures in RNA-Seq data (*Racle et al., 2017*; *Sturm et al., 2019*). We normalize the ATAC-Seq counts

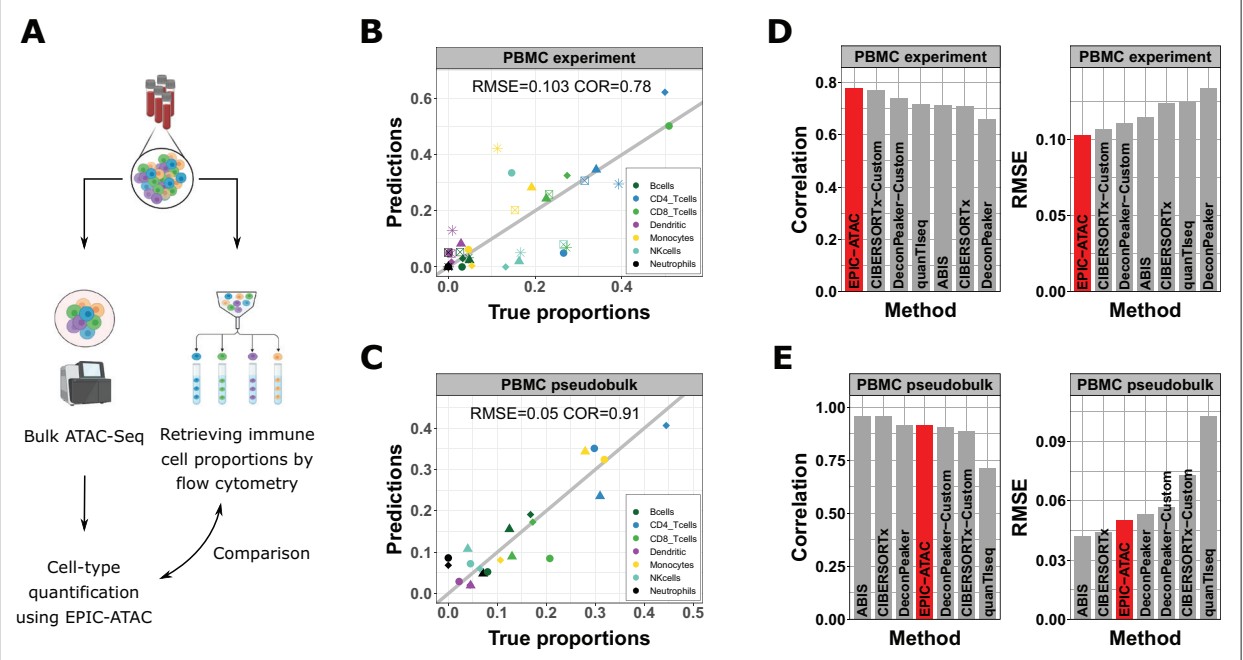

**Figure 3.** EPIC-ATAC accurately estimates immune cell fractions in PBMC ATAC-Seq samples. (**A**) Schematic description of the experiment designed to validate the ATAC-Seq deconvolution on PBMC samples. Created with BioRender.com. (**B**) Comparison between cell-type proportions predicted by EPIC-ATAC and the true proportions in the PBMC bulk dataset. Symbols correspond to donors. (**C**) Comparison between the proportions of cell types predicted by EPIC-ATAC and the true proportions in the PBMC pseudobulk dataset. Symbols correspond to pseudobulks. (**D**) Pearson correlation (left) and RMSE (right) values obtained by each deconvolution tool on the PBMC bulk dataset. The EPIC-ATAC results are highlighted in red. (**E**) Pearson correlation (left) and RMSE (right) values obtained by each deconvolution tool on the PBMC pseudobulk dataset.

The online version of this article includes the following figure supplement(s) for figure 3:

**Figure supplement 1.** Cellular composition of the samples from the PBMC experiment.

**Figure supplement 2.** Cell-type proportions estimated by different deconvolution methods on the PBMC datasets.

by dividing counts by the peak lengths as well as sample depth and rescaling counts so that the counts of each sample sum to $10^6$. In RNA-Seq-based deconvolution, EPIC uses an estimation of the amount of mRNA in each reference cell type to derive cell proportions while correcting for cell-type specific mRNA bias. For the ATAC-Seq-based deconvolution, these values were set to 1 to give similar weights to all cell-types quantifications. Indeed ATAC-Seq measures signal at the DNA level, hence the quantity of DNA within each reference cell type is similar.

## EPIC-ATAC accurately estimates immune cell fractions in PBMC ATAC-Seq samples

To test the accuracy of EPIC-ATAC predictions, we first collected PBMCs from five healthy donors. For each donor, half of the cells were used to generate a bulk ATAC-Seq dataset and the other half was used to determine the cellular composition of each sample, that is the proportions of monocytes, B cells, CD4+ T cells, CD8+ T cells, NK cells and dendritic cells, by multiparametric flow cytometry (*Figure 3A*, see Materials and methods). We then applied EPIC-ATAC to the bulk ATAC-Seq data. The predicted cell fractions are consistent with the cell fractions obtained by flow cytometry (*Figure 3B*, Pearson correlation coefficient of 0.78 and root mean squared error (RMSE) of 0.10). However, the accuracy of the estimations is variable depending on the samples. In particular, one sample (*Sample4*, star shape in *Figure 3B*) exhibits larger discrepancies between EPIC-ATAC predictions and the ground truth. Visualizing our marker peaks in the five PBMC samples (*Figure 3—figure supplement 1*) showed that this sample might be an outlier considering that its cellular composition is similar to that of Samples 2 and 5 but this sample shows particularly high ATAC-Seq accessibility at the monocytes and dendritic marker peaks. This could explain why EPIC-ATAC overestimates the proportions of the two populations in this case.

As a second validation, we applied EPIC-ATAC to pseudobulk PBMC samples (referred to as the PBMC pseudobulk dataset), generated using three publicly available PBMC scATAC-Seq datasets (*Granja et al., 2019*; *Satpathy et al., 2019*; *10x Genomics, 2021*; see Materials and methods). A high correlation (0.91) between EPIC-ATAC predictions and true cell-type proportions and a low RMSE (0.05) were observed for this dataset (*Figure 3C*).

The accuracy of the predictions obtained with EPIC-ATAC was then compared with the accuracy of other deconvolution approaches which could be used with our reference profiles and marker peaks (*Figure 3D–E*). To this end, we considered both the DeconPeaker method (*Li et al., 2020b*) originally developed for bulk ATAC-Seq as well as several algorithms developed for bulk RNA-Seq (CIBER-SORTx *Newman et al., 2019*), quanTIseq (*Finotello et al., 2019*), ABIS (*Monaco et al., 2019*), and MCPcounter (*Becht et al., 2016*). To enable meaningful comparison across the cell types considered in this work and use the method initially developed for bulk RNA-Seq deconvolution, the marker peaks and profiles derived in this work were used in each of these methods (See Materials and methods, 'Benchmarking of the EPIC-ATAC framework against other existing deconvolution tools'). As in EPIC-ATAC, the features of the input bulk matrices were matched to our reference profiles' features. Having reference profiles/markers and an ATAC-Seq bulk query with matched features was the only requirement of the different deconvolution models to run on ATAC-Seq data with the default methods parameters. DeconPeaker and CIBERSORTx also include the option to define cell-type specific markers and profiles from a set of reference samples. We thus additionally fed our ATAC-Seq samples collection to both algorithms and used the resulting profiles and marker peaks to perform bulk ATAC-Seq deconvolution. The resulting predictions are referred to as DeconPeaker-Custom and CIBERSORTx-Custom.

Many tools displayed high correlation and low RMSE values, similar to those of EPIC-ATAC, and no single tool consistently outperformed the others (*Figure 3D–E*, *Figure 3—figure supplement 2*). The fact that our marker peaks and reference profiles could be used with EPIC-ATAC and other existing tools demonstrates their broad applicability.

## EPIC-ATAC accurately predicts fractions of cancer and non-malignant cells in tumor samples

We evaluated the ability of the EPIC-ATAC framework to predict not only immune and stromal cell proportions but also the proportion of cells for which reference profiles are not available (i.e. uncharacterized cells). For this purpose, we considered two previously published scATAC-Seq datasets containing basal cell carcinoma and gynecological cancer samples (*Satpathy et al., 2019*; *Regner et al., 2021*) as well as the samples from the Human Tumor Atlas Network (HTAN; https://data.humantumoratlas.org/) single-cell multiome dataset composed of samples from diverse cancer types (*Terekhanova et al., 2023*). For each dataset, we generated pseudobulks by averaging the chromatin accessibility signal across all cells of each sample (see Materials and methods, 'Datasets used for the evaluation of ATAC-Seq deconvolution'). Applying EPIC-ATAC to each dataset shows that this framework can simultaneously predict the proportions of both uncharacterized cells and immune, stromal and vascular cells (*Figure 4A*). In basal cell carcinoma and gynecological cancer samples, the proportion of uncharacterized cells can be seen as a proxy for the proportion of cancer cells. In the HTAN dataset, some of the samples also contain cell types that are neither immune, fibroblasts or endothelial cells nor malignant cells. Hence, the uncharacterized cells in these samples group cancer and normal cells from the tumor site. EPIC-ATAC, in this context, was able to not only estimate the proportion of cell types included in the TME reference profiles but also the proportion of uncharacterized cells in most of the cancer types (*Figure 4A*, *Figure 4—figure supplement 1*).

As for the PBMC datasets, we compared EPIC-ATAC performances to other existing deconvolution tools (see Materials and methods, 'Benchmarking of the EPIC-ATAC framework against other existing deconvolution tools'). For each dataset, EPIC-ATAC led to the highest performances and was the only method to accurately predict the proportion of uncharacterized cells (*Figure 4B*, *Figure 4—figure supplements 2 and 3* and *Figure 4—figure supplement 4*). Although quanTIseq also allows users to perform such predictions, the method resulted in lower correlation and higher RMSE values when comparing the estimated and true proportions of the uncharacterized cells (*Figure 4B*, *Figure 4—figure supplements 2 and 4*).

In the EPIC-ATAC and quanTIseq frameworks, predictions correspond to absolute cell-type fractions which include the proportion of uncharacterized cells, that is proportions of all cells present in

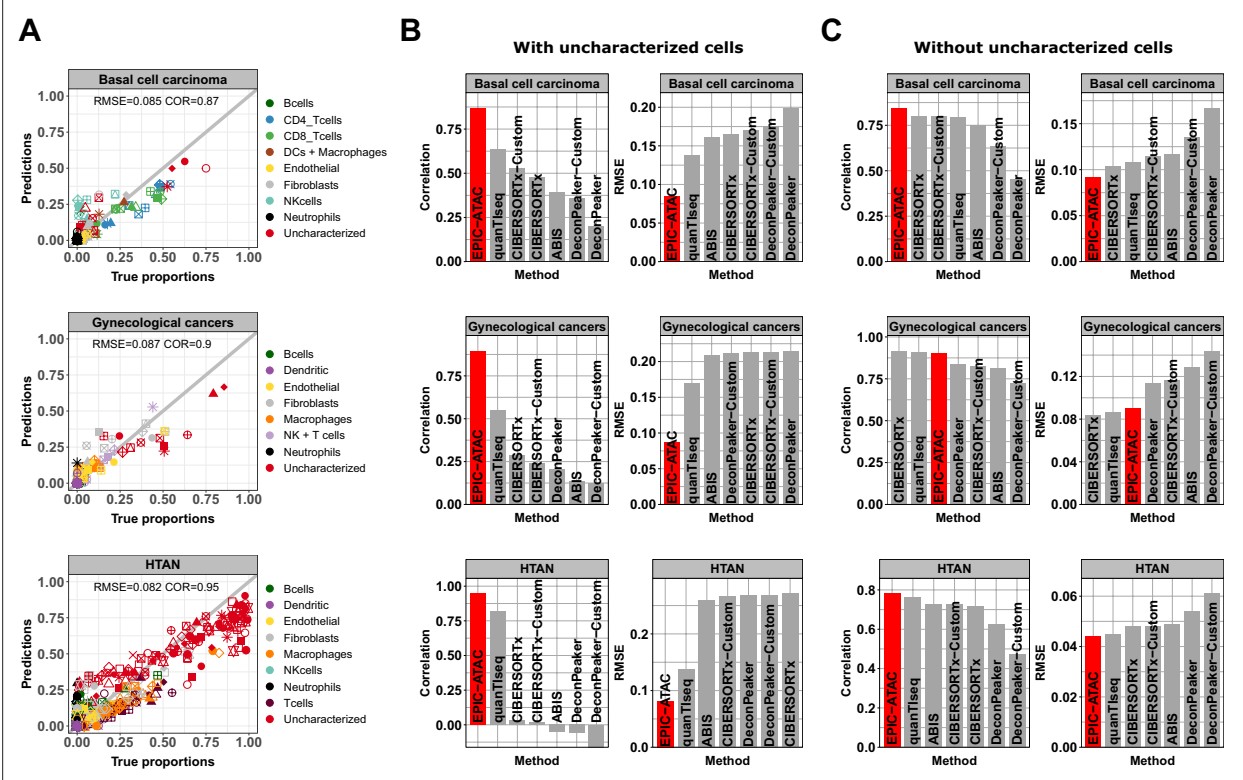

**Figure 4.** EPIC-ATAC accurately predicts fractions of cancer and non-malignant cells in tumor samples. (**A**) Comparison between cell-type proportions estimated by EPIC-ATAC and true proportions for the basal cell carcinoma (top), gynecological (middle) and HTAN (bottom) pseudobulk datasets. Symbols correspond to pseudobulks. (**B**) Pearson's correlation and RMSE values obtained for the deconvolution tools included in the benchmark. EPIC-ATAC is highlighted in red. (**C**) Same analyses as in panel B, with the uncharacterized cell population excluded for the evaluation of the prediction accuracy. The predicted and true proportions of the immune, stromal and vascular cell types were rescaled to sum to 1.

The online version of this article includes the following figure supplement(s) for figure 4:

**Figure supplement 1.** EPIC-ATAC estimations across different cancer types in the HTAN data.

**Figure supplement 2.** Cell-type proportions estimated by different deconvolution methods on the tumor samples.

**Figure supplement 3.** Cell-type proportions estimated by CIBERSORTx and DeconPeaker in tumor samples, using reference profiles built from our collection of pure ATAC-Seq samples.

**Figure supplement 4.** RMSE values for cell-type predictions obtained by each tool on the tumor samples.

**Figure supplement 5.** CPU time in seconds needed to run each tool on the benchmarking datasets.

the bulk, while the estimations obtained from the other tools correspond to relative cell fractions, that is proportions of cells present in the reference profiles (CIBERSORTx, DeconPeaker) or to scores with arbitrary units (ABIS, MCPcounter) without considering the presence of uncharacterized cells. For the latter group of tools, we fixed the uncharacterized cells estimations to 0. This approach provides a clear and significant advantage to EPIC-ATAC and quanTIseq (*Figure 4B*). For this reason, wconducted a second benchmark excluding the predictions of uncharacterized cell fractions and rescaling both estimations and true proportions to sum to 1 (see Materials and methods, 'Benchmarking of the EPIC-ATAC framework against other existing deconvolution tools'). EPIC-ATAC outperformed most of the other methods also when excluding the uncharacterized cells (*Figure 4C*, *Figure 4—figure supplements 2–4*). Note that in terms of computational resources, all the tools require a limited amount of time and memory to run, that is less than 20 s in average (*Figure 4—figure supplement 5*).

## Accuracy of ATAC-Seq deconvolution is determined by the abundance and specificity of each cell type

To investigate the impact of cell type abundance on the accuracy of ATAC-Seq deconvolution, we evaluated EPIC-ATAC predictions in each major cell type separately in the different benchmarking datasets

(*Figure 5A*). NK cells, endothelial cells, neutrophils, or dendritic cells showed lower correlation values. These values can be explained by the fact that these cell types are low-abundant in our benchmarking datasets (*Figure 5A*). For the endothelial cells and dendritic cells, the RMSE values associated with these cell types remain low. This suggests that while the predictions of EPIC-ATAC might not be precise enough to compare these cell-type proportions between different samples, the cell-type quantification within each sample is reliable. For the NK cells and the neutrophils, we observed more variability with higher RMSE values in some datasets which suggests that the markers and profiles for these cell types might be improved. *Figure 5—figure supplements 1 and 2* detail the performances of each tool when considering each cell type separately in the PBMC and the cancer datasets. As for EPIC-ATAC, the predictions from the other deconvolution tools are more reliable for the frequent cell types.

To explore the accuracy of ATAC-Seq deconvolution for more related cell types, we evaluated whether EPIC-ATAC could predict the proportions of T-cell subtypes. To this end, we considered naive and non-naive CD8+ as well as naive, helper/memory and regulatory CD4+ T cells (Tregs). We redefined our list of cell-type specific marker peaks and reference profiles including also these five T-cell subtypes (*Supplementary files 8-9*, *Figure 5—figure supplement 3A*) and observed that the markers were conserved in external data (*Figure 5—figure supplement 3B*). The annotations of the markers associated with the T-cell subtypes are available in *Supplementary files 10-13*.

We capitalized on the more detailed cell-type annotation of the PBMC datasets as well as the basal cell carcinoma dataset to evaluate the EPIC-ATAC predictions of cell-subtype fractions using these updated markers and profiles. Overall, the correlations observed between the predictions and true proportions of T cells decreased when considering T-cell subtypes rather than CD4+ and CD8+ cell types only (*Figure 5B*). In particular, low accuracies were obtained for helper/memory CD4+ and naive T-cell subtypes (*Figure 5C*). Similar results were obtained using other deconvolution tools (*Figure 5—figure supplement 4*).

## EPIC-ATAC accurately infers the immune contexture in a bulk ATAC-Seq breast cancer cohort

We applied EPIC-ATAC to a breast cancer cohort of 42 breast ATAC-Seq samples including samples from two breast cancer subtypes, that is 35 oestrogen receptor (ER)-positive human epidermal growth factor receptor 2 (HER2)-negative (ER+/HER2-) breast cancer samples and 7 triple negative breast cancer (TNBC) samples (*Kumegawa et al., 2023*). No cell sorting was performed in parallel to the chromatin accessibility sequencing. For this reason, the authors used a set of cell-type specific cis-regulatory elements (CREs) identified in scATAC-Seq data from similar breast cancer samples (*Kumegawa et al., 2022*) and estimated the amount of infiltration of each cell type by averaging the ATAC-Seq signal of each set of cell-type specific CREs in their samples. We used EPIC-ATAC to estimate the proportions of different cell types of the TME. These predictions were then compared to the metric used by Kumegawa and colleagues in their study to infer levels of infiltration. A high correlation between the two metrics was observed for each cell type (Pearson's correlation coefficient from 0.5 for myeloid cells to 0.94 for T cells, *Figure 6A*).

We observed a higher proportion of T cells, B cells, and NK cells in the TNBC samples in comparison to ER+/HER2- samples (*Figure 6B*). We then compared the cellular composition of ER+/HER2- subgroups identified in the original study (clusters CA-A, CA-B and CA-C). A higher infiltration of T and B cells was observed in cluster CA-C and higher proportions of endothelial cells and fibroblasts were observed in cluster CA-B (*Figure 6C*). These differences recapitulate those reported in the Kumegawa study except for the difference in myeloid infiltration observed in the original study between the different breast cancer subgroups (*Kumegawa et al., 2023*). However, when considering each myeloid cell type present in our reference profiles separately, a higher infiltration of macrophages was observed in the TNBC samples in comparison to the ER+/HER2- samples (*Figure 6—figure supplement 1*). Also, we observed a difference in the levels of NK cell infiltration between TNBC and ER+/HER2- samples while no NK infiltration estimation was provided in the original paper for this cell type.

## EPIC-ATAC and EPIC RNA-seq based deconvolution have similar accuracy and can complement each other

We compared the accuracy of EPIC when applied to ATAC-Seq data and to RNA-Seq data. For this purpose, we first used single-cell multiomic data which provides for each cell both its chromatin

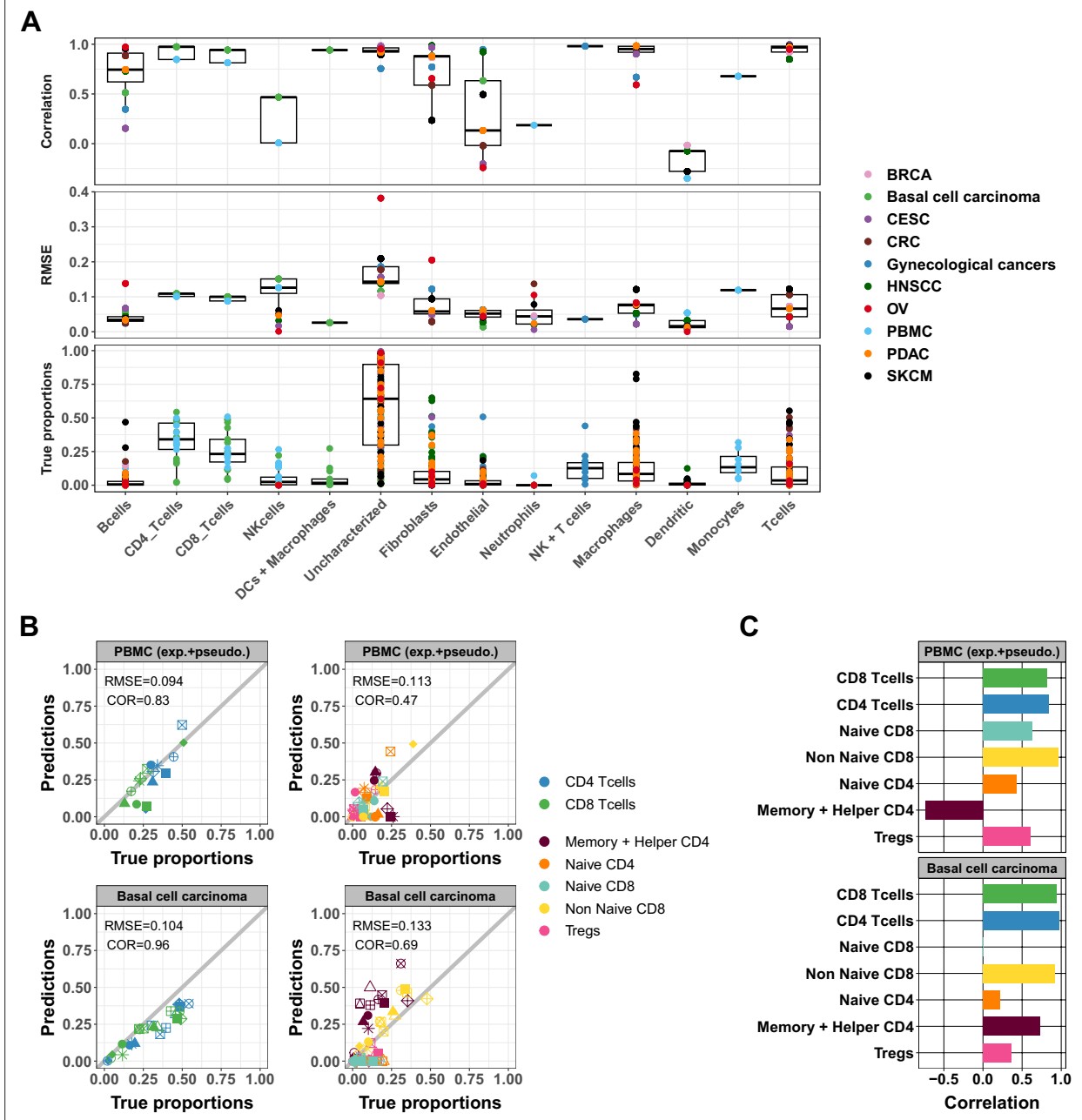

**Figure 5.** Accuracy of ATAC-Seq deconvolution is determined by the abundance and specificity of each cell type. (**A**) Correlations (top) and RMSE (middle) between EPIC-ATAC predictions and true cell-type proportions in each cell type. True proportions are also shown for each cell type (bottom). Colors correspond to different datasets. (**B**) Comparison of the proportions estimated by EPIC-ATAC and the true proportions for PBMC samples (PBMC experiment and PBMC pseudobulk samples combined) (top) and the basal cell carcinoma pseudobulks (bottom). Predictions of the proportions of CD4+ and CD8+ T cells were obtained using the reference profiles based on the major cell types, and subtypes predictions using the reference profiles including the T-cell subtypes. (**C**) Pearson's correlation values obtained by EPIC-ATAC in each cell type.

The online version of this article includes the following figure supplement(s) for figure 5:

**Figure supplement 1.** Pearson's correlation and RMSE values for cell-type predictions obtained by each tool for each cell type in the PBMC samples.

**Figure supplement 2.** Pearson's correlation and RMSE values for cell-type predictions obtained by each tool for each cell type in the tumor samples.

**Figure supplement 3.** Scaled average chromatin accessibility of each cell-type-specific marker peak, including T-cell subtype markers, in the reference and validation samples.

**Figure supplement 4.** Predicted proportions of CD4+ and CD8+ T-cell subtypes using different deconvolution tools.

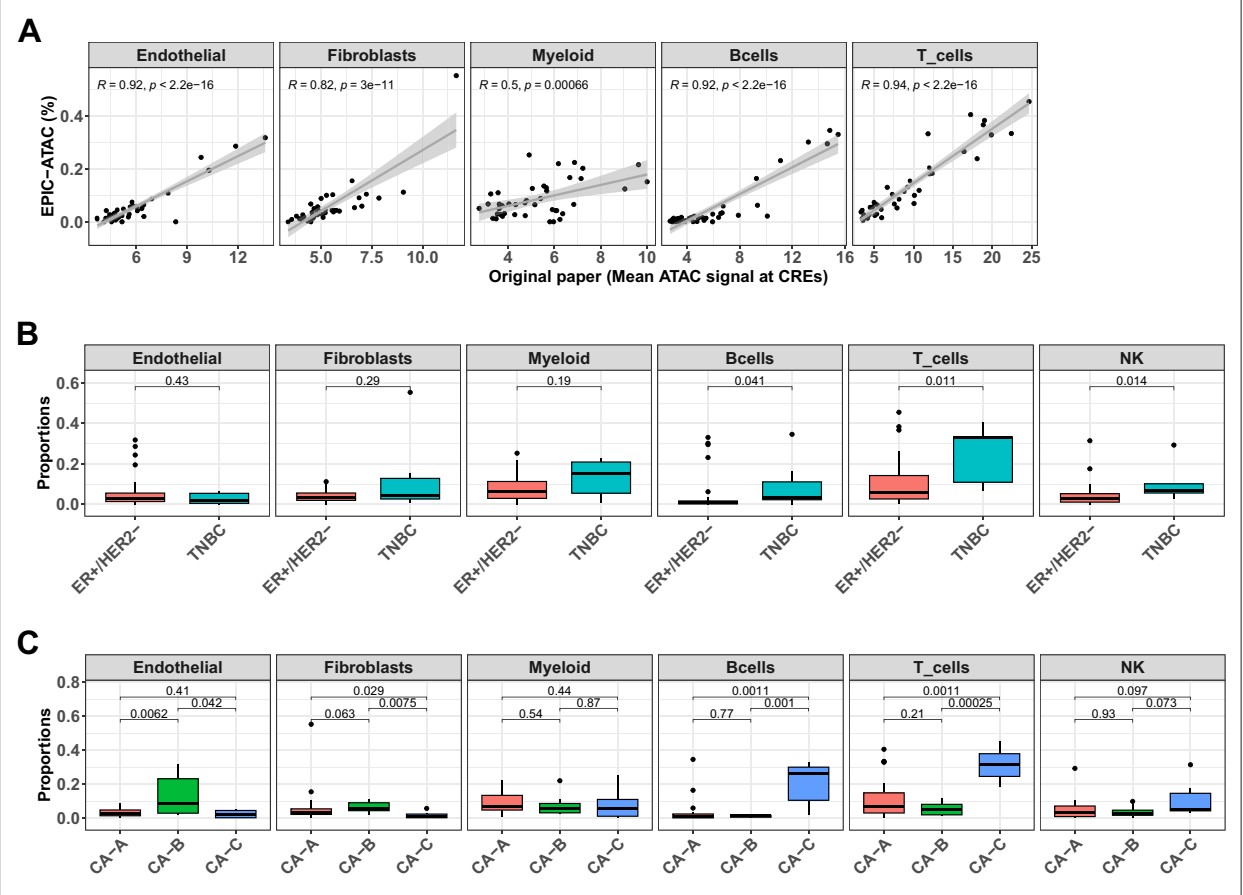

**Figure 6.** EPIC-ATAC accurately infers the immune contexture in a bulk ATAC-Seq breast cancer cohort. (**A**) Proportions of different cell types predicted by EPIC-ATAC in each sample as a function of the average ATAC signal at the cell-type specific CREs used by *Kumegawa et al., 2023* to infer the level of cell-type infiltration in the tumor samples (42 samples). Pearson's correlation coefficients and linear regression lines were calculated for each cell type, 95% confidence intervals are represented by the shaded areas. (**B**) Proportions of different cell types predicted by EPIC-ATAC in the samples stratified based on two breast cancer subtypes. (**C**) Proportions of different cell types predicted by EPIC-ATAC in the samples stratified based on three ER+/HER2- subgroups. Wilcoxon test p-values are represented at the top of the boxplots.

The online version of this article includes the following figure supplement(s) for figure 6:

**Figure supplement 1.** EPIC-ATAC predictions of the proportions of each myeloid cell type in breast cancer samples.

accessibility profile and its gene expression profile. These data were retrieved from the 10 x multiome PBMC dataset (*10x Genomics, 2021*) and the HTAN dataset (*Terekhanova et al., 2023*) (see Materials and methods, 'Datasets used for the evaluation of ATAC-Seq deconvolution'). We used EPIC-ATAC to perform ATAC-Seq-based deconvolution on the chromatin accessibility levels of the peaks and the original EPIC tool to perform standard RNA-seq deconvolution on the gene expression levels. ATAC-Seq peaks can also be aggregated into gene activity (GA) variables as a proxy for gene expression, based on peak distances to each gene. We applied the GA transformation to the pseudobulk data and performed GA-based RNA deconvolution using the original EPIC tool (See Materials and methods 'Comparing deconvolution based on RNA-Seq, gene activity or peak features'). *Figure 7A and B* show that EPIC-ATAC performs similarly to the EPIC RNA-seq-based deconvolution and outperforms the GA-based RNA deconvolution. The lower performances of GA-based RNA deconvolution could be explained by the fact that GA features, by construction, do not perfectly match the transcriptomic data.

We also applied EPIC-ATAC and the original version of EPIC (EPIC-RNA) on a bulk dataset composed of PBMC samples with matched bulk RNA-Seq, bulk ATAC-Seq and flow cytometry data (*Morandini et al., 2024*; *Figure 7C*). We compared the predictions obtained using each modality to the flow cytometry cell-type quantifications. EPIC-ATAC predictions were better correlated with

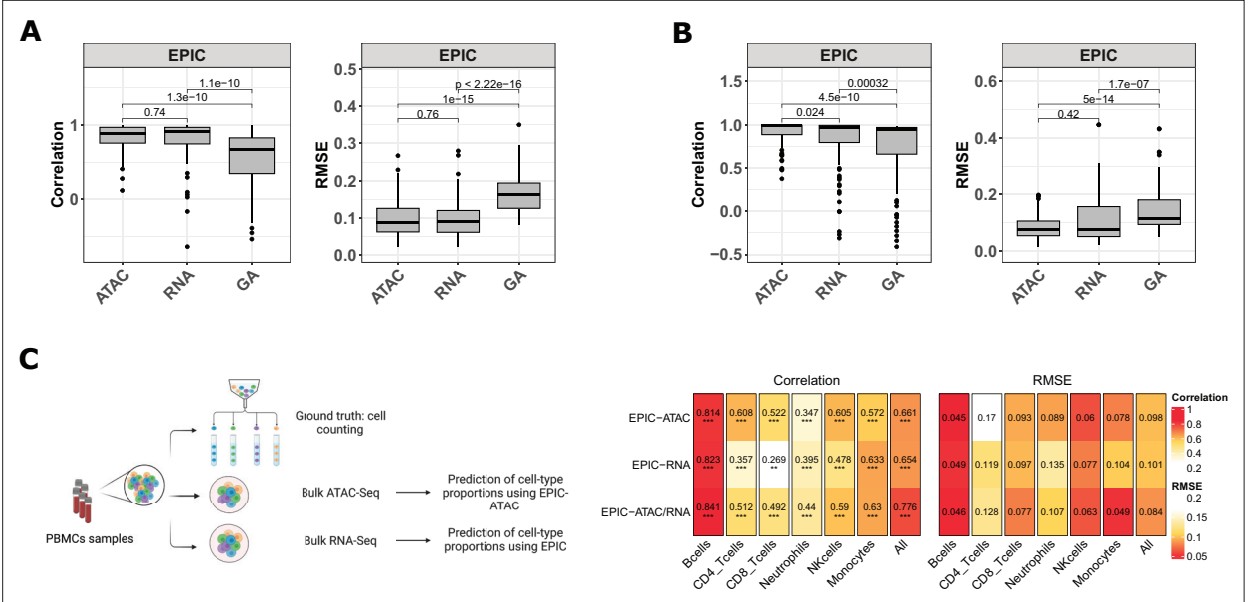

**Figure 7.** EPIC-ATAC and EPIC RNA-seq based deconvolution have similar accuracy and can complement each other. (**A–B**) Pearson's correlation (left) and RMSE (right) values comparing the proportions predicted by the ATAC-Seq deconvolution, the RNA-Seq deconvolution and the GA-based RNA deconvolution and true cell-type proportions in 100 pseudobulks simulated form the 10 x multiome PBMC dataset (**10x Genomics, 2021**) (panel **A**) and in the pseudobulks generated from the HTAN cohorts (panel **B**). Dots correspond to outlier pseudobulks. (**C**) Left panel: Schematic description of the dataset from **Morandini et al., 2024** composed of matched bulk ATAC-Seq, RNA-Seq and flow cytometry data from PBMC samples. Created with BioRender.com. Right panel: Pearson's correlation (left) and RMSE values (right) obtained by EPIC-ATAC, EPIC-RNA and EPIC-ATAC/RNA (i.e., averaged predictions of EPIC-ATAC and EPIC-RNA) in each cell type separately or all cell types together (columns).

the flow cytometry measures for some cell types (e.g. CD8+, CD4+ T cells, NK cells) while this trend was observed with the EPIC-RNA predictions in other cell types (B cells, neutrophils, monocytes; *Figure 7C*). We then tested whether the predictions obtained from both modalities could be combined to improve the accuracy of each cell-type quantification. Averaging the predictions obtained from both modalities shows a moderate improvement (*Figure 7C*), suggesting that the two modalities can complement each other.

## Discussion

Bulk chromatin accessibility profiling of biological tissues like tumors represents a reliable and affordable technology to map the activity of gene regulatory elements across multiple samples in different conditions. Here, we collected ATAC-Seq data from pure cell populations covering major immune and non-immune cell types found in the tumor microenvironment. This enabled us to identify reliable cell-type specific marker peaks and chromatin accessibility profiles for both PBMC and solid tumor sample deconvolution. We integrated these data in the EPIC deconvolution framework to accurately predict the fractions of both malignant and non-malignant cell types from bulk tumor ATAC-Seq samples.

Although not tested in this work, the TME marker peaks and profiles could be used on normal tissues where immune cells are expected to be present. In cases where specific cell types are expected in a sample but are not part of our list of reference profiles (e.g., neuronal cells in brain tumors or tissues other than human PBMCs or tumor samples), custom marker peaks and reference profiles can be provided to EPIC-ATAC to perform cell-type deconvolution. To this end, users should select markers that are cell-type specific, which could be identified using pairwise differential analysis performed on ATAC-Seq data from sorted cells from the populations of interest, following the approach developed in this work (*Figure 1*, see Code availability).

Solid tumors contain large and heterogeneous fractions of cancer cells for which it is challenging to build reference profiles. We show that the EPIC-ATAC framework, in contrast to other existing tools, allows users to accurately predict the proportion of cells not included in the reference profiles (*Figure 4* and *Figure 4—figure supplement 1*). These uncharacterized cells can include cancer cells

but also other non-malignant cells. Since the major cell types composing TMEs were included in our reference profiles, the proportion of uncharacterized cells approximates the proportion of cancer cells in most cases.

For our benchmarking, we provided our reference profiles and markers to each tool that do not provide the option to automatically build new profiles from a set of reference samples. While this allowed us to run deconvolution tools previously developed on RNA-Seq data with their default parameters, we cannot exclude that hyperparametrization could improve the performance of these tools, and we anticipate that the cell-type specific ATAC-Seq profiles and markers curated in this work could help improving bulk ATAC-Seq deconvolution with different tools. Also, for RNA-Seq data deconvolution, some of the methods followed specific pre-processing steps, for example the quan-TIseq framework removes a manually curated list of noisy genes as well as aberrant immune genes identified in the TCGA data, and ABIS uses immune-specific housekeeping genes. Additional filtering could be explored to improve the performance of deconvolution tools in the context of ATAC-Seq data.

The pseudobulk approach provides unique opportunities to design benchmarks with known cell-type proportions but also comes with some limitations. Indeed, pseudobulks are generated from single-cell data which are noisy and whose cell-type annotation is challenging in particular for closely related cell types. These limitations might lead to chromatin accessibility profiles that deviate from true bulk data and errors in the true cell-type proportions. The evaluation of our method on true bulk ATAC-Seq samples from PBMCs and breast cancer samples supported the accuracy of EPIC-ATAC to deconvolve bulk ATAC-Seq data. In the breast cancer application, the observation of similar immune compositions in TNBC and ER+/HER2- samples as the ones identified in the original paper (*Figure 6*) highlighted that the use of scATAC-Seq data, which are not always available for all cancer types, could be avoided for the estimation of different cell-type infiltration in bulk samples.

The evaluation of the EPIC-ATAC deconvolution resulted in an average absolute error of 8% across cell types. This number is consistent with previous observations in RNA-Seq data deconvolution (*Racle et al., 2017*). Considering this uncertainty, the quantification of low-frequency populations remains challenging (*Jin and Liu, 2021*). Comparing such estimations across samples should be performed with care due to the uncertainty of the predictions (*Figure 5A*).

Another limitation of cell-type deconvolution is often reached when closely related cell types are considered. In the reference-based methods used in this study, this limit was reached when considering T-cell subtypes in the reference profiles (*Figure 5B–C* and *Figure 5—figure supplement 4*). We thus recommend to use the EPIC-ATAC framework with the markers and reference profiles based on the major cell-type populations. We additionally provide the marker peaks of the T-cell subtypes which could be used to build cell-type specific chromatin accessibility signatures or perform 'peak set enrichment analysis' similarly to gene set enrichment analysis (*Subramanian et al., 2005*). Such an application could be useful for the annotation of scATAC-Seq data, which often relies on matched RNA-Seq data and for which there is a lack of markers at the peak level (*Jiang et al., 2023*).

The comparison of EPIC-ATAC applied on ATAC-Seq data with EPIC applied on RNA-Seq data has shown that both modalities led to similar performances and that they could complement each other. Another modality that has been frequently used in the context of bulk sample deconvolution is methylation. Methylation profiling techniques such as methylation arrays are cost-effective (*Kaur et al., 2023*) and DNA methylation signal is highly cell-type specific (*Kaur et al., 2023*; *Loyfer et al., 2023*). Considering that methylation and chromatin accessibility measure different features of the epigenome, additional analyses comparing and/or complementing ATAC-seq-based deconvolution with methylation-based deconvolution could be of interest as future datasets profiling both modalities in the same samples become available.

Another possible application of our marker peaks relies on their annotation (*Figure 2G*, *Supplementary files 4-5*), which could be used to expand the list of genes and CBPs associated with each cell type or subtype. For example, the neutrophils marker peaks were enriched for motifs of TFs such as SPI1 (*Supplementary file 4*), which was not listed in the neutrophil genes in the databases used for annotation but has been reported in previous studies as involved in neutrophils development (*Watt et al., 2021*). The annotations related to the set of major cell types and T-cell subtypes are provided in *Supplementary files 4 and 5*; *Supplementary files 10 and 11*. Finally, the annotation of marker peaks highlighted pathways involved in immune responses to tumoral environments (*Figure 2G*). Examples

of these pathways are the toll-like receptor signaling pathway involved in pathogen-associated and recognition of damage-associated molecular patterns in diverse cell types including B and T cells (*Geng et al., 2010*; *Javaid and Choi, 2020*), glucan metabolic processes which are known to be related to trained immunity which can lead to anti-tumor phenotype in neutrophils (*Kalafati et al., 2020*) or the Fc-receptor signaling observed in NK cells (*Bonnema et al., 1994*; *Sanseviero, 2019*). These observations suggest that our marker peaks contain regulatory regions not only specific to cell types but also adapted to the biological context of solid tumors.

## Conclusion

In this work, we identified biologically relevant cell-type specific chromatin accessibility markers and profiles for most non-malignant cell types frequently observed in the tumor microenvironment. We capitalized on these markers and profiles to predict cell-type proportions from bulk PBMC and solid tumor ATAC-Seq data (https://github.com/GfellerLab/EPIC-ATAC, copy archived at *Racle and Gabriel, 2024*). Evaluated on diverse tissues, EPIC-ATAC shows reliable predictions of immune, stromal, vascular, and cancer cell proportions. With the expected increase of ATAC-Seq studies in cancer, the EPIC-ATAC framework will enable researchers to deconvolve bulk ATAC-Seq data from tumor samples to support the analysis of regulatory processes underlying tumor development and correlate the TME composition with clinical variables.

# Materials and methods
## Generation of an ATAC-Seq reference dataset of non-malignant cell types frequently observed in the tumor microenvironment
### Pre-processing of the sorted ATAC-Seq datasets

We collected pure ATAC-Seq samples from 12 studies. The data include samples from (i) ten major immune, stromal and vascular cell types: B (*Corces et al., 2016*; *Calderon et al., 2019*; *Zhang et al., 2022a*), CD4+ (*Corces et al., 2016*; *Mumbach et al., 2017*; *Liu et al., 2020*; *Giles et al., 2022*; *Zhang et al., 2022a*), CD8+ (*Corces et al., 2016*; *Calderon et al., 2019*; *Liu et al., 2020*; *Giles et al., 2022*; *Zhang et al., 2022a*), natural killer (NK) (*Corces et al., 2016*; *Calderon et al., 2019*), dendritic (DCs) cells (*Calderon et al., 2019*; *Leylek et al., 2020*; *Liu et al., 2020*), macrophages (*Liu et al., 2020*; *Zhang et al., 2022a*), monocytes (*Corces et al., 2016*; *Calderon et al., 2019*; *Leylek et al., 2020*; *Trizzino et al., 2021*; *Zhang et al., 2022a*) and neutrophils (*Perez et al., 2020*; *Ram-Mohan et al., 2021*) as well as fibroblasts (*Liu et al., 2020*; *Ge et al., 2021*) and endothelial (*Liu et al., 2020*; *Xin et al., 2020*) cells (See *Figure 2A*), and (ii) eight tissues from distinct organs (i.e. bladder, breast, colon, liver, lung, ovary, pancreas and thyroid) from the ENCODE data (*Abascal et al., 2020*; *Rozowsky et al., 2023*). The list of the samples and their associated metadata (including cell types and accession number of the study of origin) is provided in *Supplementary file 1*. To limit batch effects, the samples were reprocessed homogeneously from the raw data (fastq files) processing to the peak calling. For that purpose, raw fastq files were collected from GEO using the SRA toolkit and the PEPATAC framework (*Smith et al., 2021*) was used to process the raw fastq files based on the following tools: trimmomatic for adapter trimming, bowtie2 (with the PEPATAC default parameters) for reads pre-alignment on human repeats and human mitochondrial reference genome, bowtie2 (with the default PEPATAC parameters: `--very-sensitive` -*X 2000*) for alignment on the human genome (hg38), samtools (PEPATAC default parameters: -*q 10*) for duplicates removal and MACS2 (*Zhang et al., 2008*) (PEPATAC default parameters: `--shift -75 --extsize 150 --nomodel --call-summits --nolambda --keep-dup` *all* -*p 0.01*) for peak calling in each sample. After alignment, reads mapping on chromosome M were excluded. TSS enrichment scores were computed for each sample and used to filter out samples with low quality (criteria of exclusion: TSS score <5) (See *Supplementary file 1* containing the TSS score of each sample). 789 samples (including 564 from our 10 reference cell types) had a TSS score >5.

### Generation of a consensus set of peaks

Peak calling was performed in each sample individually. Peaks were then iteratively collapsed to generate a set of reproducible peaks. For each cell type, peak collapse was performed adapting the iterative overlap peak merging approach proposed in the PEPATAC framework. A first peak collapse

was performed at the level of each study of origin, that is, if peaks identified in distinct samples overlapped (minimum overlap of 1 bp between peaks), only the peak with the highest peak calling score was kept. Also, only peaks detected in at least half of the samples of each study were considered for the next step. If a study had only two samples, only peaks detected in both samples were considered. After this first selection, a second round of peak collapse was performed at the cell-type level to limit batch effects in downstream analyses. For each cell type, only peaks detected in all the studies of origin were considered. The final list of peaks was then generated by merging each set of reproducible peaks. Peaks located on chromosome Y were excluded from the rest of the analyses. ATAC-Seq counts were retrieved for each sample and each peak using *featureCounts* (*Liao et al., 2014*).

## Identification of cell-type specific markers

### Differential accessibility analysis

To identify cell-type specific markers, we split the samples collection into ten folds (created with the *create_folds* function from the R package splitTools *Mayer, 2023*). For each fold, we performed pairwise differential accessibility analysis across the ten cell types considered in the reference samples as well as the ENCODE samples from diverse organs. The differential analysis was performed using limma (*Ritchie et al., 2015*, version 3.56.2). Effective library sizes were computed using the method of trimmed mean of M-values (TMM) from the edgeR package in R (*Robinson et al., 2010*, version 3.42.4). Due to differences in library size across all samples collected, we used voom from the limma package (*Law et al., 2014*) to transform the data and model the mean-variance relationship. Finally, a linear model was fitted to the data to assess the differential accessibility of each peak across each pair of cell types. To identify our marker peaks, all peaks with log2 fold change higher than 0.2 were selected and ranked by their maximum adjusted *p*-value across all pairwise comparisons. The top 200 features (with the lowest maximum adjusted *p*-value) were considered as cell-type specific marker peaks. The marker peaks identified in at least three folds were considered in the final list of marker peaks.

### Marker peaks filtering

Modules of open chromatin regions accessible in all (universal modules) or in specific human tissues have been identified in the study *Zhang et al., 2021b*. These regions were used to refine the set of marker peaks and exclude peaks with residual accessibility in other cell types than those considered for deconvolution. More precisely, for immune cells, endothelial and fibroblasts specific peaks, we filtered out the peaks overlapping the universal modules as well as the tissue specific modules except for the immune (modules 8–25), endothelial (modules 26–35) and stromal related modules (modules 41–49 and 139–150) respectively. As a second filtering step, we retained markers exhibiting the highest correlation patterns in tumor bulk samples from different solid cancer types, that is The Cancer Genome Atlas (TCGA) samples (*Corces et al., 2018*). We used the Cancer Genomics Cloud (CGC) (*Lau et al., 2017*) to retrieve the ATAC-Seq counts for each marker peak in each TCGA sample (using *featureCounts*). For each set of cell-type specific peaks, we identified the most correlated peaks using the *findCorrelation* function of the caret R package (*Kuhn, 2008*, version 6.0–94) with a correlation cutoff value corresponding to the 90th percentile of pairwise Pearson correlation values.

### Evaluation of the study of origin batch effect

To identify potential batch effect issues, we run principal component analysis (PCA) based on the cell-type specific peaks after normalizing ATAC-Seq counts using full quantile normalization (FQ-FQ) implemented in the EDASeq R package (*Risso et al., 2011*) to correct for depth and GC biases. These data were used to visualize the samples in three dimensions using the first axes of the PCA based on the PBMC and TME markers (*Figure 2B* and *Figure 2—figure supplement 1*). We also used the ten first principal components to evaluate distances between samples and compute silhouette coefficients based on the cell-type and study of origin classifications as well as to compute the ARI metric comparing the cell-type annotation and the clustering obtained using model-based clustering.

## Building the reference profiles

It has been previously demonstrated in the context of RNA-Seq-based deconvolution approaches (*Racle et al., 2017*; *Sturm et al., 2019*) that the transcripts per million (TPM) transformation is appropriate to estimate cell fractions from bulk mixtures. We thus normalized the ATAC-Seq counts of the reference samples using a TPM-like transformation, that is dividing counts by peak length, correcting sample counts for depth and rescaling counts so that the counts of each sample sum to $10^6$. We then computed for each peak the median of the TPM-like counts across all samples from each cell type to build the reference profiles of the 10 cell types considered in the EPIC-ATAC framework (*Figure 2C*). In the EPIC algorithm, weights reflecting the variability of each feature of the reference profile can be considered in the constrained least square optimization. We thus also computed the interquartile range of the TPM-like counts for each feature in each cell type. Two ATAC-Seq reference profiles are available in the EPIC-ATAC framework: (i) a reference profile containing profiles for B cells, CD4+ T cells, CD8+ T cells, NK, monocytes, dendritic cells and neutrophils to deconvolve PBMC samples, and (ii) a reference profile containing profiles for B cells, CD4+ T cells, CD8+ T cells, NK, dendritic cells, macrophages, neutrophils, fibroblasts and endothelial cells to deconvolve tumor samples. The reference profiles are available in the EPICATAC R package and the reference profiles restricted to our cell-type specific marker peaks are available in *Supplementary files 2 and 3*.

## Assessing the reproducibility of the marker peaks signal in independent samples

We evaluated the chromatin accessibility level of the marker peaks in samples that were not included in the peak calling step. Firstly, we considered samples from two independent studies (*Ucar et al., 2017*; *Carvalho et al., 2021*) providing pure ATAC-Seq data for five immune cell types (i.e. B, CD4+ T cells, CD8+ T cells, Monocytes, Macrophages) (*Figure 2D*). To consider the other cell types, samples that were excluded from the reference dataset due to a low TSS enrichment score were also considered in this validation dataset (*Supplementary file 1*). Secondly, we collected the data from a single-cell atlas chromatin accessibility from human tissues and considered the cell types included in our reference data (*Zhang et al., 2021b*; *Figure 2E*). We used the cell-type annotations provided in the original study (GEO accession number: GSE184462). The Signac R package (*Stuart et al., 2021*, 1.9.0) was used to extract fragment counts for each cell and each marker peak and the ATAC-Seq signal of each marker peak was averaged across all cells of each cell type.

## Annotation of the marker peaks

The cell-type specific markers were annotated using ChIPseeker R package (*Yu et al., 2015*, version 1.34.1) and the annotation from *TxDb.Hsapiens.UCSC.hg38.knownGene* in R to identify the regions in which the marker peaks are (i.e. promoter, intronic regions, etc.) and ChIP-Enrich to associate each peak to the nearest gene TSS (*Welch et al., 2014*). The nearest genes identified were then compared to cell-type marker genes listed in the PanglaoDB (*Franzén et al., 2019*) and CellMarker databases (*Hu et al., 2023*). PanglaoDB provides an online interface to explore a large collection on single-cell RNA-Seq data as well as a community-curated list of cell-type marker genes. CellMarker is a database providing a large set of curated cell-type markers for more than 400 cell types in human tissues retrieved from a large collection of single-cell studies and flow cytometry, immunostaining or experimental studies. ChIP-Enrich was also used to perform gene set enrichment and identify for each set of cell-type specific peaks potential biological pathways regulated by the marker peaks. The enrichment analysis was performed using the *chipenrich* function (*genesets = "GOBP"*, *locusdef = "nearest_tss"*) from the chipenrich R package (v2.22.0).

Chromatin accessibility peaks can also be annotated for chromatin binding proteins (CBPs) such as transcription factors (TFs), whose potential binding in the peak region is reported in databases. In our study, we chose the JASPAR2022 (*Castro-Mondragon et al., 2022*) database and the ReMap database (*Hammal et al., 2022*).

Using the JASPAR2022 database, we assessed, for each cell type, whether the cell-type specific marker peaks were enriched in specific TFs motifs using two TFs enrichment analysis frameworks: Signac (*Stuart et al., 2021*) and MonaLisa (*Machlab et al., 2022*). For the MonaLisa analysis, the cell-type specific markers peaks were categorized in bins of sequences, one bin per cell type (use of the *calcBinnedMotifEnrR* function). To test for enrichment of motifs, the sequences of each bin were

compared to a set of background peaks with similar average size and GC composition obtained by randomly sampling regions in all the peaks identified from the reference dataset. The enrichment test was based on a binomial test. For the Signac analysis, we used the *FindMotif* function to identify over-represented TF motifs in each set of cell-type specific marker peaks (query). This function used a hypergeometric test to compare the number of query peaks containing the motif with the total number of peaks containing the motif in the background regions (matched to have similar GC content, region length and dinucleotide frequencies as the query regions), corresponding in our case to the peaks called in the reference dataset.

The ReMap database associates chromatin binding proteins (CBPs), including TFs, transcriptional coactivators and chromatin-remodeling factors, to their DNA binding regions based on DNA-binding experiments such as chromatin immunoprecipitation followed by sequencing (ChIP-seq). For each association of a CBP to its binding region, the cell type in which the binding has been observed is reported in the ReMap database (biotype). We used the ReMapEnrich R package (version 0.99) to test if the cell-type specific marker peaks are significantly enriched in CBPs-binding regions listed in the *Hammal et al., 2022* catalog. We considered the non-redundant peaks catalog from ReMmap 2022, containing non-redundant binding regions for each CBP in each biotype. Similarly to the previously mentioned enrichment methods, we chose the consensus peaks called in the reference samples as the universe for the enrichment test. Note that, for each cell type, an enrichment was retained only if the biotype in which the CBP-regions were identified matched the correct cell type.

## Datasets used for the evaluation of ATAC-Seq deconvolution

### PBMCs ATAC-Seq data from healthy donors

#### Peripheral blood mononuclear cell (PBMC) isolation

Venous blood from five healthy donors was collected at the local blood transfusion center of Geneva in Switzerland, under the approval of the Geneva University Hospital's Institute Review Board, upon written informed consent and in accordance with the Declaration of Helsinki. Volunteers were asked to read an information sheet for blood donation and to complete an online medical questionnaire on the day of donation. After the questionnaire was finalized, a PDF file was generated for printing and signed to give approval for blood donation. PBMCs were freshly isolated by Lymphoprep (Promega) centrifugation (1800 rpm, 20 min, without break, room temperature). Red blood cell lysis was performed using red blood lysis buffer (QIAGEN) and platelets were removed by centrifugation (1000 rpm, 10 min without break, room temperature). Cells were counted and immediately used.

#### Flow cytometry

Immune cell populations were identified using multiparameter flow cytometry and the following antibodies: FITC anti-human CD45RA (HI100, Biolegend), PerCP-Cyanine5.5 anti-human CD19 (H1B19, Biolegend), PE anti-human CD3 (SK7, Biolegend), PE-Dazzle anti-human CD14 (M0P9, BD Biosciences), PE-Cyanine7 anti-human CD56 (HCD56, Biolegend), APC anti-human CD4 (RPA-T4, Biolgend), APC-Cyanine7 anti-human CCR7 (G043H7, Biolegend), Brilliant Violet 421 anti-human CD8 (RPA-T8, Biolegend), Brilliant Violet 510 anti-human CD25 (BC96, Biolegend), Brilliant Violet 711 anti-human CD16 (3G8, Biolegend), Brilliant Violet 786 anti-human CD127 (A019D5, Biolegend), Ultra-Brilliant Violet anti-human CD45 (HI30, BD Biosciences), FITC anti-human Celc9a (8F9, Miltenyi), PE anti-human XCR1 (S15046E, Biolegend), PE-Dazzle anti-human BDCA-2 (201 A, Biolegend), APC anti-human BDCA-3 (AD5-14H12, Miltenyi), Brilliant Violet 421 anti-human CD3 (UCHT1, Biolegend), Brilliant Violet 421 anti-human CD14 (M5E2, BD Pharmingen), Brilliant Violet 421 anti-human CD19 (SJ25C1, Biolegend), Brilliant Violet 510 anti-human BDCA-1 (L161, Biolegend), Brilliant Violet 650 anti-human CD11c (3.9, Biolegend), Brilliant Violet 711 anti-human CD11c (N418, Biolegend) and Brilliant Violet 711 anti-human HLA-DR (L243, Biolegend). Dead cells were excluded using the Zombie UV Fixable Viability Kit (Biolegend). Intracellular staining was performed after fixation and permeabilization of the cells with the FoxP3 Transcription Factor Staining Buffer Set (00-5523-00, Invitrogen) using Alexa 700 anti-human FoxP3 antibody (259D/C7, BD Biosciences). Data were acquired on LSRFortessa flow cytometer and analysed using FlowJo software (v10.7.1).

## Cell preparation for ATAC-sequencing

50,000 CD45+ cells were sorted from total PBMCs using anti-human Ultra-Brilliant Violet (BUV395) CD45 (HI30, BD Biosciences) with a FACSAria II (Becton Dickinson) and were collected in PBS with 10% Foetal Bovine Serum (FBS). Cell pellets were resuspended in cold lysis buffer (10 mM Tris-Cl pH 7.4, 10 mM NaCl, 3 mM MgCl2, 0,1% NP40 and water) and immediately centrifuged at 600 × g for 30 min at 4 °C. Transposition reaction was performed using the Illumina Tagment DNA Enzyme and Buffer kit (20034210, Illumina) and transposed DNA was eluted using the MinElute PCR Purification Kit (QIAGEN). Libraries were generated by PCR amplification using indexing primers and NEBNext High-Fidelity Master Mix (New England BioLabs) and were purified using AMPure XP beads (A63880, Beckman Coulter). Libraries were quantified by a fluorometric method (QubIT, Life Technologies) and their quality assessed on a Fragment Analyzer (Agilent Technologies). Sequencing was performed as a paired-end 50 cycles run on an Illumina NovaSeq 6000 (v1.5 reagents) at the Genomic Technologies Facility (GTF) in Lausanne, Switzerland. Raw sequencing data were demultiplexed using the bcl2fastq2 Conversion Software (version 2.20, Illumina).

## Data processing

The same steps as for the processing of the reference ATAC-Seq samples were followed. (See Pre-processing of the ATAC-Seq datasets).

## ATAC-Seq pseudobulk data from PBMCs and cancer samples

To evaluate the accuracy of our ATAC-Seq deconvolution framework, we generated pseudo-bulk datasets from 6 single-cell datasets:

- **PBMC pseudobulk dataset:** a combination of three single-cell datasets for PBMCs.
  - Dataset 1 corresponds to a scATAC-Seq dataset obtained from *Satpathy et al., 2019* (GEO accession number: GSE129785). This dataset contains FACS-sorted populations of PBMCs. Since the cells of some cell types came from a unique donor, all the cells of this dataset were aggregated to form one pseudobulk. Ground truth cell fractions were obtained by dividing the number of cells in each cell type by the total number of cells.
  - Dataset 2 (included in the PBMC pseudobulk dataset) was retrieved from *Granja et al., 2019* (GEO accession number GSE139369). B cells, monocytes, dendritic, CD8+, CD4+ T, NK cells, neutrophils from healthy donors were considered. The neutrophil cells came from a single donor. As for dataset 1, we thus aggregated all the cells to generate one pseudobulk. Ground truth cell fractions were obtained by dividing the number of cells in each cell type by the total number of cells.
  - Dataset 3 (included in the PBMC pseudobulk dataset) corresponds to the 10 x multiome dataset of PBMC cells (*10x Genomics, 2021*). Since these data come from one donor, one pseudobulk sample was generated for this dataset. The pseudobulk was generated by averaging the ATAC-Seq signal from all cells from the following cell types: B cells, CD4+ T cells, CD8+ T cells, NK cells, Dendritic cells, and monocytes.
- **Basal cell carcinoma dataset:** obtained from the study of *Satpathy et al., 2019*. This dataset is a scATAC-Seq dataset composed of 13 basal cell carcinoma samples composed of immune (B cells, plasma cells, CD4+ T cells, CD8+ T cells, NK cells, myeloid cells), stromal (endothelial and fibroblasts) and cancer cells. Plasma cells and cancer cells were both considered as uncharacterized cells (i.e. cell types not included in the reference profiles). Cell annotations were retrieved from the original study.
- **Gynecological cancer dataset:** obtained from the study of *Regner et al., 2021* (GEO accession number GSE173682). In this study, the authors performed scATAC-Seq on 11 gynecological cancer samples from two tumor sites (i.e. endometrium and ovary) and composed of immune (B cells, NK and T cells grouped under the same cell-type annotation, macrophages, mast cells), stromal (fibroblast, endothelial, smooth muscle) and cancer cells. Mast cells, smooth muscle and cancer cells were considered as uncharacterized cells. Cell annotations were retrieved from the original study.
- **HTAN:** obtained from the study of *Terekhanova et al., 2023* from the HTAN DCC Portal. In this study, the authors generated an atlas of single-cell multiomic data (RNA-Seq and ATAC-Seq profiling for the same cell) for diverse cancer types. We considered the annotation provided by the HTAN consortium. This cell annotation was composed of immune cells (B cells, T cells,

dendritic cells, macrophages), fibroblasts, endothelial cells, cancer cells, and normal cells. The two latter cell groups were considered as part of the uncharacterized cell population.

For Basal cell carcinoma, Gynecological cancer and the HTAN datasets, one pseudobulk per sample was generated and ground truth cell fractions were obtained for each sample by dividing the number of cells in each cell type by the total number of cells in the sample.

For each dataset, raw fragments files were downloaded from the respective GEO accession numbers and data were preprocessed using ArchR (*Granja et al., 2021*, ArchR R package 1.0.2). Cells with TSS score below four were removed. Doublets removal was performed using the *doubletsRemoval* function from ArchR. To match as much as possible real bulk ATAC-seq data processing, peak calling was not performed on each cell type or cell cluster as usually done in scATAC-Seq studies but using all cells for each dataset from the PBMC pseudobulk data or grouping cells by sample for the Basal cell carcinoma, Gynecological cancer and HTAN datasets. Peak calling was performed using MACS2 within the ArchR framework. Fragment counts were extracted using ArchR for each peak called to generate single-cell peak count matrices. Note that for the HTAN dataset, in some cancer types, less than 15 peaks were in common with our reference profiles peaks. These cancer types were thus not considered in the analysis (GBM, UCEC, CEAD). These matrices were normalized using a TPM-like transformation, *i.e.*, dividing counts by peak length and correcting sample counts for depth. Finally, for each peak, the average of the normalized counts was computed across all the cells for each dataset from the PBMC pseudobulk data and across all the cells of each sample for the Basal cell carcinoma, Gynecological cancer and HTAN datasets. Averaged data were then rescaled so that the sum of counts of each sample sum to $10^6$.

## Bulk ATAC-Seq data from a breast cancer cohort

Bulk ATAC-Seq samples from a breast cancer cohort were obtained from *Kumegawa et al., 2023*. These data include 42 breast cancer samples which can be classified based on two features: (i) the breast cancer subtype ER+/HER2- or triple negative, and (ii) the molecular classification provided by the original study (CA-A, CA-B, and CA-C). The ATAC-Seq raw counts and the sample metadata were retrieved from figshare (*Kumegawa, 2023*). As for the previously mentioned datasets, raw counts were normalized using the TPM-like transformation prior to bulk deconvolution.

## Benchmarking of the EPIC-ATAC framework against other existing deconvolution tools

### Tools included in the benchmark

The performances of the EPIC-ATAC framework were benchmarked against the following deconvolution tools:

- quanTIseq (*Finotello et al., 2019*) is a deconvolution tool using constrained least square regression to deconvolve RNA-Seq bulk samples. No reference profiles are available in this framework to perform ATAC-Seq deconvolution and quanTIseq does not provide the option to automatically build reference profiles from pure bulk samples. quanTIseq was thus run using the reference profiles derived in this work for the EPIC-ATAC framework and the *quanTIseq* function from the quantiseqr R package. Note that the correction for cell-type specific mRNA content bias was disabled for ATAC-Seq data deconvolution (parameters used in quantiseqr: scaling set to 1 for all cell types and method set to 'lsei').
- DeconPeaker (*Li et al., 2020b*) relies on SIMPLS, a variant of partial least square regression to perform bulk RNA-Seq and bulk ATAC-Seq deconvolution. ATAC-Seq reference profiles are available in this deconvolution framework however not all cell types considered in the EPIC-ATAC framework are included in the DeconPeaker reference profiles. This tool was thus run using different reference profiles: (i) the reference profiles derived in this work for the EPIC-ATAC framework (corresponds to 'DeconPeaker' or 'DeconPeaker_ourmarkers' in our analyses), and (ii) reference profiles automatically generated by DeconPeaker from the sorted reference samples collected in this work (corresponds to 'DeconPeaker_Custom' in our analyses). The results of DeconPeaker obtained using its original markers and profiles are also provided for the cell types in common with the cell types considered in this work in *Figure 5—figure supplements 1 and 2*. Deconvolution was run using the deconvolution module deconPeaker (using findctsps with the following parameter: `--lib-strategy=ATAC-Seq`). DeconPeaker outputs cell-type proportions relative to the total amount of cells from the reference cell types.

- CIBERSORTx *Newman et al., 2019* is a deconvolution algorithm based on linear support vector regression. CIBERSORTx does not provide ATAC-Seq reference profiles, however it is possible to automatically generate new profiles from a set of pure bulk samples. This tool was thus run using different reference profiles: (i) the reference profiles derived in this work for the EPIC-ATAC framework (corresponds to 'CIBERSORTx' or 'CIBERSORTx_ourMarkers' in our analyses), and (ii) reference profiles automatically generated by CIBERSORTx from the sorted reference samples collected in this work (corresponds to 'CIBERSORTx_Custom' in our analyses). To run CIBERSORTx, we used the docker container provided by the authors of CIBERSORTx on their website. The algorithm was run using the default options (i.e. `--absolute FALSE`, `--rmbatchBmode FALSE` and `-rmbatchSmode FALSE`), which results in cell-type proportions relative to the total amount of cells from the reference cell types.
- ABIS (*Monaco et al., 2019*) uses robust linear modeling to estimate cell-type proportions in bulk RNA-Seq samples. No ATAC-Seq reference profiles are available in the deconvolution framework. ABIS was run using the EPIC-ATAC reference profiles by using the *rlm* function from the MASS R package (as performed in the deconvolute_abis function from the immunedeconv R package *Sturm et al., 2019*) to quantify each cell type from the reference profiles. The cell-type quantifications returned by this approach are in arbitrary units. To compare the estimations and the true cell proportions, we scaled the estimations of each sample between 0 and 1 to obtain relative proportions.
- MCPcounter (*Becht et al., 2016*): MCPcounter returns scores instead of cell-type proportions. The scores were obtained using the *appendSignatures* function from the MCPcounter R package by providing the list of marker peaks specific to each cell type. The cell-type scores are not comparable between cell types, MCPcounter was thus included only in the evaluation of the performances in each cell type separately.

For all the tools, TPM-like data were used as input bulk samples for the deconvolution.

Since CIBERSORTx, ABIS and DeconPeaker do not predict proportions of uncharacterized cells, we performed two benchmarking analyses: (i) including all cell types and (ii) excluding the cell types that are absent from the reference profiles (uncharacterized cells) and rescaling the estimated and true proportions of the immune cells, endothelial cells and fibroblasts so that their sum equals 1.

### Benchmarking of the running time of each tool

We compared the running time of each tool on the different datasets considered in our benchmarking and used the bash command *time* to retrieve the CPU time of each run. The measured running time is composed of the following steps: (i) matching the bulk features to the reference profiles' features and (ii) running the deconvolution algorithm.

## Comparing deconvolution based on RNA-Seq, gene activity or peak features

100 pseudobulks were generated from the 10x PBMC multiome dataset (*10x Genomics, 2021*) based on 3000 cells for each pseudobulk. Cell fractions were defined using the *rdirichlet* function from the *gtools* R package. We also considered pseudobulks built from the HTAN datasets by averaging the signal in all cells belonging to a sample. Three sets of features were extracted from the data, that is gene expression features extracted from the RNA-Seq layer, ATAC-Seq peaks and gene activity derived from the ATAC-Seq layer. The same cells were considered for each modality.

Gene activity features were extracted from the single-cell data using ArchR (1.0.2), which considers distal elements and adjusts for large differences in gene size in the gene activity score calculation. Gene activity pseudobulks were built by averaging the gene activity scores across all cells belonging to the pseudobulk. For ATAC-Seq pseudobulk, peaks called using ArchR on all cells from the single-cell ATAC data were considered (see Materials and methods, 'ATAC-Seq pseudobulk data from PBMCs and cancer samples') and counts were averaged across all cells of each pseudobulk. For RNA-Seq pseudobulks, counts were also averaged across all cells of each pseudobulk. All aggregated data were depth normalized across each feature to $10^6$. Cell-type deconvolution was performed on each pseudobulk using EPIC-ATAC on the peak matrix using our ATAC-Seq marker peaks and reference profiles. The RNA-Seq and gene activity pseudobulks were deconvolved with EPIC.

Bulk ATAC-Seq samples from a PBMC cohort were also obtained from *Morandini et al., 2024*. These data include more than 100 samples with matched bulk RNA-Seq, bulk ATAC-Seq as well as

matched flow cytometry data. The ATAC-Seq and RNA-Seq raw counts were obtained from GEO using the following GEO accession numbers: GSE193140 and GSE193141. As for the previously mentioned datasets, raw counts were normalized using the TPM-like transformation prior to bulk deconvolution. The flow cytometry data were obtained from the *Supplementary file 7* from the original publication and the data were rescaled as follows: CD4+ and CD8+ T cells proportions were rescaled to sum up to the proportion of T cells and the T cells, B cells and NK cells proportions were rescaled to sum up to the proportion of lymphocytes. All proportions were then rescaled to sum up to 1. EPIC-ATAC was applied to the ATAC-Seq data and EPIC to the RNA-Seq data and the performances of each set of predictions were compared to each other.

## Code availability

The code to download and preprocess publicly available ATAC-Seq samples as well as the code used to identify our cell-type specific marker peaks, generate the reference profiles and perform the main analyses of the paper is available on GitHub (https://github.com/GfellerLab/EPIC-ATAC_manuscript, copy archived at *Gabriel, 2024*). A README file is provided on the GitHub repository with more details on how to use the code.

The code to perform ATAC-Seq deconvolution using the EPIC-ATAC framework is available as an R package called EPICATAC and is available on GitHub (https://github.com/GfellerLab/EPIC-ATAC, *Racle and Gabriel, 2024*).

## Acknowledgements

We thank the Lausanne Genomic Technologies Facility, University of Lausanne, Switzerland (https://wp.unil.ch/gtf/) for the sequencing of the PBMC samples as well as Yan Liu, Dana Moreno and Matei Teleman for testing the EPICATAC R package. Some of the illustrations were created with BioRender.com and published using a CC BY-NC-ND license with permission (*Figures 1, 3A and 7C*).

## Additional information

### Competing interests

David Gfeller: has recieved consulting fees from CeCaVa and Gnubiotics. The other authors declare that no competing interests exist.

### Funding

| Funder | Grant reference number | Author |
|---|---|---|
| University of Lausanne | | Aurélie Anne-Gaëlle Gabriel |

The funders had no role in study design, data collection and interpretation, or the decision to submit the work for publication.

### Author contributions

Aurélie Anne-Gaëlle Gabriel, Conceptualization, Data curation, Software, Formal analysis, Visualization, Methodology, Writing – original draft, Writing – review and editing; Julien Racle, Conceptualization, Software, Visualization, Methodology, Writing – review and editing; Maryline Falquet, Camilla Jandus, Resources, Writing – review and editing; David Gfeller, Conceptualization, Supervision, Funding acquisition, Visualization, Methodology, Writing – original draft, Writing – review and editing

### Author ORCIDs

Aurélie Anne-Gaëlle Gabriel https://orcid.org/0000-0002-0606-3622
Julien Racle https://orcid.org/0000-0002-0100-0323
David Gfeller https://orcid.org/0000-0002-3952-0930

### Ethics

Venous blood from five healthy donors was collected at the local blood transfusion center of Geneva in Switzerland, under the approval of the Geneva University Hospital's Institute Review Board, upon

written informed consent and in accordance with the Declaration of Helsinki. Volunteers were asked to read an information sheet for blood donation and to complete an online medical questionnaire on the day of donation. After the questionnaire was finalized, a PDF file was generated for printing and signed to give approval for blood donation.

Reviewer #1 (Public Review): https://doi.org/10.7554/eLife.94833.4.sa1
Reviewer #2 (Public Review): https://doi.org/10.7554/eLife.94833.4.sa2
Author response https://doi.org/10.7554/eLife.94833.4.sa3

---

# Additional files

## Supplementary files
- Supplementary file 1. Metadata of the ATAC-Seq samples used in the study.
- Supplementary file 2. Averaged chromatin accessibility of the PBMC marker peaks in each cell type.
- Supplementary file 3. Averaged chromatin accessibility of the TME marker peaks in each cell type.
- Supplementary file 4. Annotations of the cell-type specific PBMC marker peaks.
- Supplementary file 5. Annotations of the cell-type specific TME marker peaks.
- Supplementary file 6. GO pathways enriched in each set of cell-type specific PBMC marker peaks.
- Supplementary file 7. GO pathways enriched in each set of cell-type specific TME marker peaks.
- Supplementary file 8. Averaged chromatin accessibility of the PBMC marker peaks in each cell type (T cells subtypes included).
- Supplementary file 9. Averaged chromatin accessibility of the TME marker peaks in each cell type (T cells subtypes included).
- Supplementary file 10. Annotations of the cell-type specific PBMC marker peaks (T cells subtypes included).
- Supplementary file 11. Annotations of the cell-type specific TME marker peaks (T cells subtypes included).
- Supplementary file 12. GO pathways enriched in each set of cell-type specific PBMC marker peaks (T cell subtypes).
- Supplementary file 13. GO pathways enriched in each set of cell-type specific TME marker peaks (T cell subtypes).
- MDAR checklist

## Data availability
The newly generated ATAC-Seq data have been deposited on Zenodo (https://zenodo.org/records/13132868). The other data related to this work are available in the supplementary files and on the Zenodo deposit (https://zenodo.org/records/13132868).

The following dataset was generated:

| Author(s) | Year | Dataset title | Dataset URL | Database and Identifier |
|---|---|---|---|---|
| Gabriel AAG, Racle J, Falquet M, Jandus C, Gfeller D | 2024 | Bulk PBMC ATAC-Seq data from 5 healthy donors | https://doi.org/10.5281/zenodo.13132868 | Zenodo, 10.5281/zenodo.13132868 |

The following previously published datasets were used:

| Author(s) | Year | Dataset title | Dataset URL | Database and Identifier |
|---|---|---|---|---|
| Buenrostro J | 2016 | Lineage-specific and single-cell chromatin accessibility charts human hematopoiesis and leukemia evolution | https://www.ncbi.nlm.nih.gov/geo/query/acc.cgi?acc=GSE74912 | NCBI Gene Expression Omnibus, GSE74912 |
| Calderon D, Nguyen ML, Mezger A | 2018 | Landscape of stimulation-responsive chromatin across diverse human immune cells | https://www.ncbi.nlm.nih.gov/geo/query/acc.cgi?acc=GSE118189 | NCBI Gene Expression Omnibus, GSE118189 |
| Zhang P, Amarasinghe H, Brown A, Knight J | 2022 | Epigenomic analysis reveals a dynamic and context-specific macrophage enhancer landscape associated with innate immune activation and tolerance | https://www.ncbi.nlm.nih.gov/geo/query/acc.cgi?acc=GSE172116 | NCBI Gene Expression Omnibus, GSE172116 |
| Mumbach MR, Satpathy AT, Boyle EA, Dai C, Gowen BG, Cho S, Nguyen ML, Rubin AJ, Granja J, Kazane K, Wei Y, Nguyen T, Greenside PG, Corces MR, Tycko J, Simeonov DR, Suliman N, Li R, Xu J, Flynn RA, Kundaje A, Khavari PA, Marson A, Corn JE, Quertermous T, Greenleaf WJ, Chang HY | 2017 | Enhancer connectome in primary human cells reveals target genes of disease-associated DNA elements | https://www.ncbi.nlm.nih.gov/geo/query/acc.cgi?acc=GSE101498 | NCBI Gene Expression Omnibus, GSE101498 |
| Qu K, Liu Q, Longmire M, Zaba LC, Li R, Giresi P, Kim YH, Chang HY | 2020 | Cell-type specific epigenetic regulome in Systemic Sclerosis skin | https://www.ncbi.nlm.nih.gov/geo/query/acc.cgi?acc=GSE99702 | NCBI Gene Expression Omnibus, GSE99702 |
| Giles J, Manne S, Wherry EJ | 2022 | Human epigenetic and transcriptional T cell differentiation atlas for identifying functional T cell-specific enhancers | https://www.ncbi.nlm.nih.gov/geo/query/acc.cgi?acc=GSE179593 | NCBI Gene Expression Omnibus, GSE179593 |
| Leylek R, Idoyaga J | 2020 | ATAC-seq of human dendritic cells | https://www.ncbi.nlm.nih.gov/geo/query/acc.cgi?acc=GSE146896 | NCBI Gene Expression Omnibus, GSE146896 |
| Trizzino M, Zucco A, Deliard S, Wang F, Barbieri E, Welsh S, Veglia F, Gabrilovich D, Gardini A | 2020 | EGR1 is a gatekeeper of inflammatory enhancers in human macrophages | https://www.ncbi.nlm.nih.gov/geo/query/acc.cgi?acc=GSE136216 | NCBI Gene Expression Omnibus, GSE136216 |
| Perez C, Botta C, Zabaleta A, Puig N | 2020 | Immunogenomic identification and characterization of granulocytic myeloid derived suppressor cells in multiple myeloma | https://www.ncbi.nlm.nih.gov/geo/query/acc.cgi?acc=GSE150023 | NCBI Gene Expression Omnibus, GSE150023 |
| Yang S, Ram Mohan N, Thair S | 2020 | Integrative profiling of early host chromatin accessibility responses in human neutrophils with sensitive pathogen detection | https://www.ncbi.nlm.nih.gov/geo/query/acc.cgi?acc=GSE153520 | NCBI Gene Expression Omnibus, GSE153520 |

*Continued on next page*

*Continued*

| Author(s) | Year | Dataset title | Dataset URL | Database and Identifier |
|---|---|---|---|---|
| Ge X, Frank-Bertoncelj M, Klein K, Mcgovern A, Kuret T, Houtman M, Burja B, Micheroli R, Marks M, Filer A, Buckley CD, Orozco G, Distler O, Morris AP, Martin P, Eyre S, Ospelt C | 2020 | Functional genomics atlas of synovial fibroblasts defining rheumatoid arthritis heritability | https://www.ncbi.nlm.nih.gov/geo/query/acc.cgi?acc=GSE163548 | NCBI Gene Expression Omnibus, GSE163548 |
| Su B, Wang Y, Xin J, Zhang H, He Y | 2020 | Multi-omics HUVEC data of Tibetan and Han in hypoxic induction | https://www.ncbi.nlm.nih.gov/geo/query/acc.cgi?acc=GSE145774 | NCBI Gene Expression Omnibus, GSE145774 |
| The ENCODE Consortium | 2014 | Production ENCODE epigenomic data | https://www.ebi.ac.uk/ena/browser/view/PRJNA63443 | European Nucleotide Archive, PRJNA63443 |
| Ucar D, Márquez EJ, Chung CH, Marches R, Rossi RJ, Uyar A, Wu TC, George J, Stitzel ML, Palucka AK, Kuchel GA, Banchereau J | 2017 | The chromatin accessibility signature of human immune aging stems from CD8+ T cells | https://ega-archive.org/studies/EGAS00001002605 | European Genome-Phenome Archive, EGAS00001002605 |
| Carvalho K, Mortazavi SA | 2021 | Uncovering the Gene Regulatory Networks Underlying Macrophage Polarization Through Comparative Analysis of Bulk and Single-Cell Data | https://www.ncbi.nlm.nih.gov/geo/query/acc.cgi?acc=GSE164498 | NCBI Gene Expression Omnibus, GSE164498 |
| Zhang K, Hocker JD, Miller M, Hou X, Poirion OB, Wang A, Preissl S, Ren B | 2021 | A single-cell atlas of chromatin accessibility in the human genome | https://www.ncbi.nlm.nih.gov/geo/query/acc.cgi?acc=GSE184462 | NCBI Gene Expression Omnibus, GSE184462 |
| Granja J, Zheng G, Shah P | 2019 | Massively parallel single-cell chromatin landscapes of human immune cell development and intratumoral T cell exhaustion | https://www.ncbi.nlm.nih.gov/geo/query/acc.cgi?acc=GSE129785 | NCBI Gene Expression Omnibus, GSE129785 |
| Granja JM | 2019 | Single-cell, multi-omic analysis identifies regulatory programs in mixed phenotype acute leukemia | https://www.ncbi.nlm.nih.gov/geo/query/acc.cgi?acc=GSE139369 | NCBI Gene Expression Omnibus, GSE139369 |
| 10x Genomics | 2021 | PBMC from a Healthy Donor - Granulocytes Removed Through Cell Sorting (10k) | https://www.10xgenomics.com/datasets/pbmc-from-a-healthy-donor-granulocytes-removed-through-cell-sorting-10-k-1-standard-2-0-0 | 10x Genomics, pbmc-from-a-healthy-donor-granulocytes-removed-through-cell-sorting-10-k-1-standard-2-0-0 |
| Regner MJ, Wisniewska K, Franco HL | 2021 | A multi-omic single-cell atlas of human gynecological malignancies | https://www.ncbi.nlm.nih.gov/geo/query/acc.cgi?acc=GSE173682 | NCBI Gene Expression Omnibus, GSE173682 |

*Continued on next page*

*Continued*

| Author(s) | Year | Dataset title | Dataset URL | Database and Identifier |
|---|---|---|---|---|
| Kumegawa K | 2023 | ATAC-seq data of 42 BC samples as SummarizedExperiment object with count matrix, normalized count matrix, peak info, and clinical info in Kumegawa K et al., British Journal of Cancer, 2023 | https://doi.org/10. 6084/m9.figshare. 21992609.v1 | figshare, 10.6084/ m9.figshare.21992609.v1 |
| Rechsteiner C, Morandini F, Perez K, Ocampo A | 2022 | Development of a novel aging clock based on chromatin accessibility [ATAC-seq] | https://www.ncbi. nlm.nih.gov/geo/ query/acc.cgi?acc= GSE193140 | NCBI Gene Expression Omnibus, GSE193140 |
| Rechsteiner C, Morandini F, Perez K, Ocampo A | 2022 | Development of a novel aging clock based on chromatin accessibility [RNA-seq] | https://www.ncbi. nlm.nih.gov/geo/ query/acc.cgi?acc= GSE193141 | NCBI Gene Expression Omnibus, GSE193141 |

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
