## [Editor Report · eLife assessment]

This study presents an **important** computational tool for the quantification of the cellular composition of human tissues profiled with ATAC-seq. The methodology and its application results on breast cancer tumor tissues are **convincing**. It advances existing methods by utilizing a comprehensive reference profile for major cancer-relevant cell types, compatible with a widely-used cell type deconvolution tool.

---

## [Referee Report · Reviewer #1 (Public Review)]

Summary:

Building upon their famous tool for the deconvolution of human transcriptomics data (EPIC), Gabriel et al. implemented a new methodology for the quantification of the cellular composition of samples profiled with Assay for Transposase-Accessible Chromatin sequencing (ATAC-seq). To build a signature for ATAC-seq deconvolution, they first created a compendium of ATAC-seq data and derived chromatin accessibility marker peaks and reference profiles for 12 cell types, encompassing immune cells, endothelial cells, and fibroblasts. Then, they coupled this novel signature with the EPIC deconvolution framework based on constrained least-square regression to derive a dedicated tool called EPIC-ATAC. The method was then assessed using real and pseudo-bulk ATAC-seq data from human peripheral blood mononuclear cells (PBMC) and, finally, applied to ATAC-seq data from breast cancer tumors to show it accurately quantifies their immune contexture.

Strengths:

Overall, the work is of very high quality. The proposed tool is timely; its implementation, characterization, and validation are based on rigorous methodologies and results in robust estimates. The newly-generated, validation data and the code are publicly available and well-documented. Therefore, I believe this work and the associated resources will greatly benefit the scientific community.

Weaknesses:

In the benchmarking analysis, EPIC-ATAC was compared also to deconvolution methods that were originally developed for transcriptomics and not for ATAC-seq data. However, the authors described in detail the specific settings used to analyze this different data modality as robustly as possible, and they discussed possible limitations and ideas for future improvement.

---

## [Referee Report · Reviewer #2 (Public Review)]

Summary:

The manuscript expands the current bulk sequencing data deconvolution toolkit to include ATAC-seq. The EPIC-ATAC tool successfully predicts accurate proportions of immune cells in bulk tumour samples and EPIC-ATAC seems to perform well in benchmarking analyses. The authors achieve their aim of developing a new bulk ATAC-seq deconvolution tool.

Strengths:

The manuscript describes simple and understandable experiments to demonstrate the accuracy of EPIC-ATAC. They have also been incredibly thorough with their reference dataset collections and have been robust in their benchmarking endeavours and measured EPIC-ATAC against multiple datasets and tools. This tool will be valuable to the community it serves.

---

## [Author Response]

The following is the authors’ response to the previous reviews.

**Recommendations for the authors:**

**Reviewer #1 (Recommendations For The Authors):**
I praise the authors for their impressive work; all my major concerns have been addressed. I believe the revised article is much stronger and will surely raise the interest of a broad readership.I list in the following a few minor points that the authors might want to consider when finalizing the work:- It might be helpful for the reader to know if EPIC-ATAC can also be used on tissues different from tumors and PBMC/blood, and how (i.e. which reference should they use).

We thank the reviewer for this comment. In the discussion, we have clarified this point as follows:

“Although not tested in this work, the TME marker peaks and profiles could be used on normal tissues where immune cells are expected to be present. In cases where specific cell types are expected in a sample but are not part of our list of reference profiles (e.g., neuronal cells in brain tumors or tissues other than human PBMCs or tumor samples), custom marker peaks and reference profiles can be provided to EPIC-ATAC to perform cell-type deconvolution. To this end, users should select markers that are cell-type specific, which could be identified using pairwise differential analysis performed on ATAC-Seq data from sorted cells from the populations of interest, following the approach developed in this work (Figure 1, see Code availability).”

- In Fig 2 the numbers are hard to read as they are too close or overlapping.

We have updated Figure 2 to avoid the overlap between the numbers.

- In Fig 5 I see some squared around the sub-panels, but it might be due to the PDF compression.

We do not see these squares on the Figure 5 but have seen such squares on Figure 1. We have checked that all the PDF files uploaded on the eLife submission system do not contain the previously mentioned squares.

- In the Introduction, some "deconvolution concepts" are introduced (e.g. Line 63-65), but not explained/illustrated. It might be helpful to refer to a "didactic" review.

We have added two references to these sentences in the introduction:

“As described in more details elsewhere (Avila Cobos et al., 2018; Sturm et al., 2019), many of these tools model bulk data as a mixture of reference profiles either coming from purified cell populations or inferred from single-cell genomic data for each cell type.”